# Multi-omic analysis of bat versus human fibroblasts reveals altered central metabolism

N Suhas Jagannathan[1,2†], Javier Yu Peng Koh[1†], Younghwan Lee[1], Radoslaw Mikolaj Sobota[3], Aaron T Irving[4,5], Lin-fa Wang[6], Yoko Itahana[1]*, Koji Itahana[1]*, Lisa Tucker-Kellogg[1,2]*

[1]Cancer and Stem Cell Biology Programme, Duke-NUS Medical School, Singapore, Singapore; [2]Centre for Computational Biology, Duke-NUS Medical School, Singapore, Singapore; [3]Functional Proteomics Laboratory, Institute of Molecular and Cell Biology (IMCB), Agency for Science, Technology and Research, Singapore, Singapore; [4]Programme in Emerging Infectious Diseases, Duke-NUS Medical School, Singapore, Singapore; [5]Zhejiang University-University of Edinburgh Institute, Zhejiang University School of Medicine, Zhejiang University, Haining, China; [6]SingHealth Duke-NUS Global Health Institute, Singapore, Singapore

*For correspondence:
yoko.itahana@duke-nus.edu.sg (YI);
koji.itahana@duke-nus.edu.sg (KI);
tuckerNUS@gmail.com (LT-K)

†These authors contributed equally to this work

Competing interest: The authors declare that no competing interests exist.

**Abstract** Bats have unique characteristics compared to other mammals, including increased longevity and higher resistance to cancer and infectious disease. While previous studies have analyzed the metabolic requirements for flight, it is still unclear how bat metabolism supports these unique features, and no study has integrated metabolomics, transcriptomics, and proteomics to characterize bat metabolism. In this work, we performed a multi-omics data analysis using a computational model of metabolic fluxes to identify fundamental differences in central metabolism between primary lung fibroblast cell lines from the black flying fox fruit bat (*Pteropus alecto*) and human. Bat cells showed higher expression levels of Complex I components of electron transport chain (ETC), but, remarkably, a lower rate of oxygen consumption. Computational modeling interpreted these results as indicating that Complex II activity may be low or reversed, similar to an ischemic state. An ischemic-like state of bats was also supported by decreased levels of central metabolites and increased ratios of succinate to fumarate in bat cells. Ischemic states tend to produce reactive oxygen species (ROS), which would be incompatible with the longevity of bats. However, bat cells had higher antioxidant reservoirs (higher total glutathione and higher ratio of NADPH to NADP) despite higher mitochondrial ROS levels. In addition, bat cells were more resistant to glucose deprivation and had increased resistance to ferroptosis, one of the characteristics of which is oxidative stress. Thus, our studies revealed distinct differences in the ETC regulation and metabolic stress responses between human and bat cells.

## Editor's evaluation

This study analyzed the metabolism of bat cells versus human cells through a comprehensive multi-omics approach, focusing on the black flying fox fruit bat. Findings revealed that bat cells have higher expression levels of Complex I in the electron transport chain but a lower oxygen consumption rate, suggesting a unique metabolic state similar to ischemia. Despite higher levels of mitochondrial reactive oxygen species, bat cells displayed greater antioxidant reserves and resilience to metabolic stress, including glucose deprivation and ferroptosis, highlighting fundamental metabolic

differences supporting bats' increased longevity and disease resistance. The study is compelling and provides solid evidence to back the hypothesis.

## Introduction

Bats display many characteristics that set them apart from other mammals, including the capacity for wing-powered flight, low rates of cancer incidence (*Seluanov et al., 2018*), high longevity quotient (*Austad and Fischer, 1991*; *Austad, 2010*; *Wilkinson and South, 2002*), and ability to carry many viruses (as a reservoir) without ill health (*Wang et al., 2011*). Each of these traits has distinct metabolic requirements and can affect overall metabolic activity. For example, flight is an energy-intensive process that requires high metabolic rates and ATP production (*Thomas and Suthers, 1972*; *Maina, 2000*). Resistance to cancer depends on multiple factors such as reactive oxygen species (ROS) management, DNA repair (*Huang et al., 2016*; *Huang et al., 2019*), and the ability to efflux genotoxic compounds (*Koh et al., 2019*), all of which have metabolic underpinnings. Longevity and disease resistance both require tolerance to metabolic and oxidative stresses, and the ability to dampen inflammasome activation (*Ahn et al., 2019*; *Kacprzyk et al., 2017*).

Multiple studies have documented individual aspects of metabolic regulation in bats. Metabolic rates of bats during flight (ATP production) have been documented to be approximately three times greater than basal metabolic rates in other mammals of similar size (*Thomas and Suthers, 1972*). In most other mammalian species, increased ATP production also creates increased production of ROS in the mitochondria, which can eventually lead to DNA and cellular damage (*Buffenstein et al., 2008*), and activate inflammasome responses. On the other hand, low amounts of ROS are a part of homeostasis and have been linked to beneficial effects on survival in multiple contexts (*Mittler, 2017*; *Di Meo et al., 2016*). It is conceivable that the homeostatic amount of ROS for a healthy cell is species-specific, as different species may have different ways of coping with the adverse effects of ROS accumulation, e.g., ROS-induced DNA damage and lipid peroxidation, or may activate downstream pathways such as inflammasomes at different levels of ROS. Hence, for bats to be able to have a high metabolic rate without concomitant cellular/DNA damage would require one of the following to be true – improved decoupling of ATP production from ROS production (reduced leakage of electrons, so less ROS is produced), improved antioxidant defense to neutralize generated ROS, or improved repair of damage resulting in fewer deleterious effects of ROS (e.g. less inflammasome activation).

Different studies have proposed different mechanisms of ROS tolerance or antioxidant defense in bats including lower hydrogen peroxide production (*Brunet-Rossinni, 2004*; *Brunet-Rossinni and Austad, 2004*; *Podlutsky et al., 2005*; *Ungvari et al., 2008*), improved DNA repair (*Foley et al., 2018*), higher expression of heat shock proteins (*Chionh et al., 2019*) and/or a drug efflux factor, ABCB1 (*Koh et al., 2019*), and positive selection for efficient mitochondria. While the reasons could be multifactorial, no studies have performed a systems-wide comparison of mitochondrial metabolism between bats and higher mammals such as humans. Such characterization of the basal energy metabolism of bats and how it is different from other mammals such as humans could shed light on mitochondrial activity/ROS management and hint toward metabolic factors that underlie/support desirable traits in bats, e.g., longevity, low cancer/mutation rates, and disease tolerance. Toward this goal, we compared the basal metabolism of cells from black flying fox fruit bat *Pteropus alecto* (*P. alecto*) and human cells. *P. alecto* is a member of the Pteropodidae family and is among the largest fructivore bats in the world. It has a lifespan of over 20 years and is documented to co-exist with lethal zoonotic viruses like Hendra and Nipah viruses. Using a primary lung fibroblast cell line that our group had established earlier from *P. alecto* (PaLung), we conducted a comparison between PaLung cells and human primary fibroblasts WI-38 cells to elucidate fundamental metabolic differences in the mitochondria.

Since bats and humans are very different species, it is likely that data from any one high-throughput platform (e.g. transcriptomics) would show many differences. Hence, to identify consistent differences in the metabolic regulation of *P. alecto* and humans, we looked for concordance between multiple high-throughput platforms: whole-cell transcriptomics, mitochondrial proteomics, and whole-cell metabolomics. To integrate the different omics results, we use constraint-based flux sampling, which is a computational modeling technique that simulates metabolic flux patterns using existing knowledge about metabolic network connectivity and topology. Constraint-based flux modeling has been

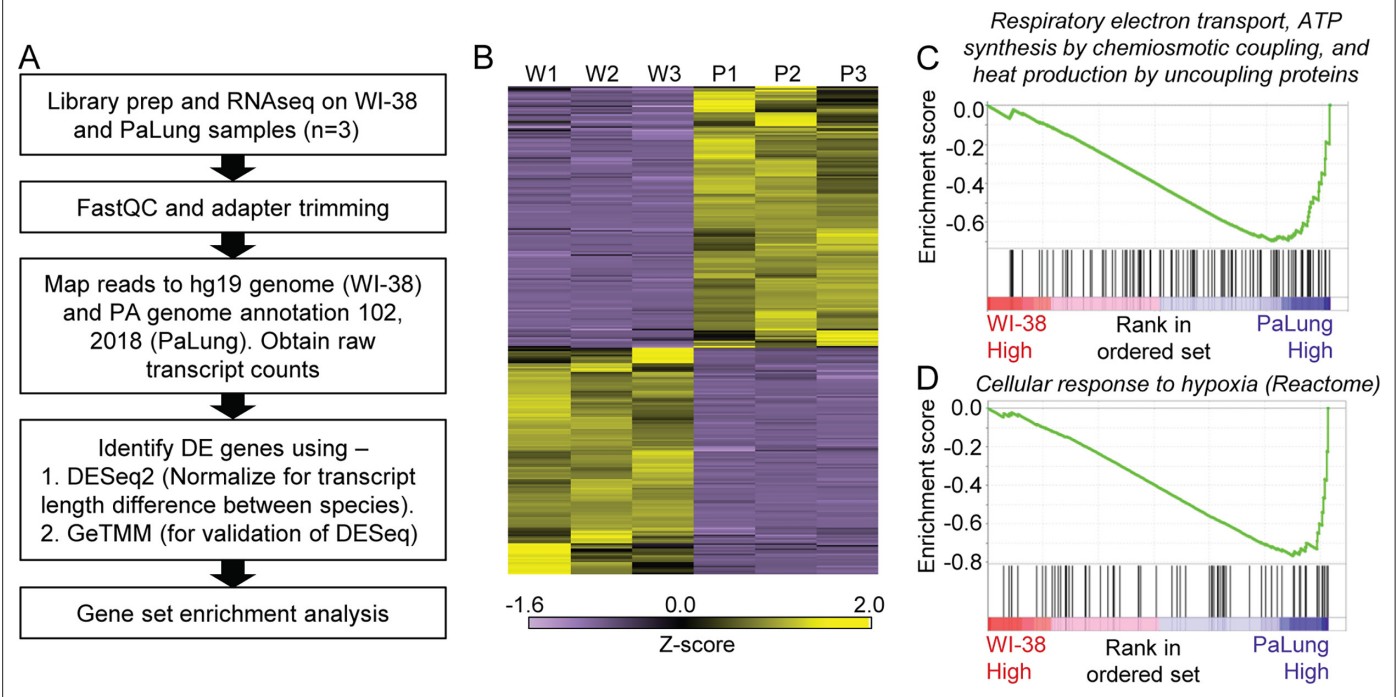

**Figure 1.** RNAseq data analysis of PaLung and WI-38 cells for differential expression and pathway enrichment. (**A**) Workflow of bioinformatics analysis pipeline for RNAseq data from PaLung (*P. alecto*) and WI-38 (*H. sapiens*) cells (*n*=3). (**B**) Heatmap showing the expression patterns for genes that passed our differential expression thresholds in the three WI-38 samples (**W1–W3**) and the three PaLung samples (**P1–P3**). (**C and D**) Gene set enrichment analysis (GSEA) identifies respiratory electron transport and cellular response to hypoxia as top metabolic pathways that are differentially regulated between PaLung and WI-38 cells. Shown here are the enrichment score plots for (**C**) respiratory electron transport and (**D**) cellular response to hypoxia.

The online version of this article includes the following figure supplement(s) for figure 1:

**Figure supplement 1.** Analysis of transcriptomics data from PaLung and WI-38 cells.

**Figure supplement 2.** A summary of transcriptomics log fold changes (LFC) overlaid onto key metabolic reactions from central carbon metabolism.

used previously for comparing metabolic phenotypes across cancers (*Aurich et al., 2017*), understanding metabolic regulation of macrophage polarization (*Bordbar et al., 2012*), studying ischemia-reperfusion injury (*Chouchani et al., 2014*), characterizing microbiomes (*Jansma and El Aidy, 2021*; *Ezzamouri et al., 2023*), optimizing metabolite production (*Patil et al., 2004*), and understanding metabolic contributors of disease pathology, e.g., diabetes (*Ravi and Gunawan, 2021*).

Here, our results show that PaLung cells have differences in basal metabolism that resemble ischemia, including the possibility of low or reverse activity of Complex II in the electron transport chain (ETC). Finally, we characterized the response of bat cells to cellular stresses such as oxidative stress, nutrient deprivation, and a type of cell death related to ischemia, viz. ferroptosis, and results were consistent with our prediction of ischemic-like basal metabolism in PaLung cells.

## Results

### Transcriptomics identifies differences in oxidative phosphorylation between PaLung cells and WI-38 cells

To understand the differences in cellular-scale metabolism between PaLung cells (*P. alecto*) and WI-38 cells (*Homo sapiens*), we performed whole-cell transcriptomics on the two cell lines. Transcriptomics detected a total of 21,952 mRNA transcripts in bat PaLung cells and 58,830 transcripts in human WI-38 cells, respectively. Since the bat genome is not as well annotated as the human genome, we performed downstream differential expression (DE) analysis using the set of 14,986 common transcripts found in both PaLung and WI-38 cells. DE analysis was performed using the DEseq pipeline, which was modified to normalize for the different transcript lengths in both species (see Materials and methods, *Figure 1A*, *Figure 1—figure supplement 1A*). This method yielded a total of 6247 transcripts that

passed our cutoff thresholds (|log fold change| ≥ 1 and FDR < 0.05, *Figure 1B*, *Figure 1—figure supplement 1B*). Because there is no standard way of normalizing RNAseq data for inter-species comparison, we also repeated the analysis in the EdgeR package, using a recently published normalization method gene length corrected trimmed mean of M-values (GeTMM) (*Smid et al., 2018*). Both methods yielded very similar results with minor discrepancies (*Figure 1—figure supplement 1C*), and we chose to perform further downstream analysis using the DEseq-generated DE transcript list. A summary of transcriptomics results for core metabolic pathways can be found in *Figure 1—figure supplement 2*, and the list of transcripts that passed our DE cutoffs can be found in *Supplementary file 1*. The number of transcripts passing our DE cutoffs (6247) was extremely high, suggesting that multiple pathways are differentially regulated between the two species. We then performed gene set enrichment analysis (GSEA), searching against the Gene Ontology Biological Process (GO BP) database. *Supplementary files 2 and 3* contain the list of enriched gene sets in PaLung and WI-38 cells respectively. Since the *P. alecto* genome is less fully annotated than the human genome, pathways with incomplete annotation may be incorrectly predicted to be downregulated in PaLung cells. Hence, we only studied differentially regulated pathways that were upregulated in PaLung. When we filtered PaLung-upregulated gene sets for significance (indicated by FDR < 0.25 and normalized enrichment score |NES| > 1), and for relevance to metabolism, only 21 gene sets remained. Many were relevant to secondary metabolism or anabolic/catabolic housekeeping, five were related to the TCA cycle and electron transport (including 'ATP synthesis by chemiosmotic coupling, respiratory electron transport, and heat production by uncoupling proteins') and three were related to hypoxic stress (such as 'cellular response to hypoxia' in the Reactome Pathway database). The genes belonging to both oxidative phosphorylation (OxPhos) (*Figure 1C*) and response to hypoxia (*Figure 1D*) gene sets had increased transcriptional expression in PaLung cells. This was interesting because, conventionally, these two pathways are active under opposing conditions (high oxygen for OxPhos vs low oxygen for hypoxia), and so would not be expected to vary in concert with each other. This hinted toward non-trivial regulation of central metabolism in PaLung cells.

## Mitochondrial proteomics suggests that OxPhos is higher in PaLung cells compared to WI-38 cells

To test whether the whole-cell RNA differences were also reflected in mitochondrial composition, we performed tandem mass tag-based proteomic profiling in the mitochondrial fractions of PaLung and WI-38 cells. Profiling detected a total of 1469 proteins, that had peptides detected with high confidence in both species. There were no peptides detected in our experiment that were exclusively detected in high confidence in only one organism. Analysis using the gene ontology tool Enrichr confirmed that a majority of these proteins were likely obtained from a mitochondrial compartment (*Figure 2—figure supplement 1A*). When we performed DE analysis on the 1469 proteins (see Materials and methods, *Figure 2A*), 405 were differentially expressed between WI-38 and PaLung cells (*Figure 2—figure supplement 1B*). Of these 405 proteins, we identified 127 to be core mitochondrial proteins (as defined by MitoCarta and IMPI datasets), that were differentially expressed between WI-38 and PaLung cells (*Supplementary file 4*). *Figure 2B* shows the heatmap of row-normalized abundances of the 127 DE proteins. We observed that most of these 127 DE mitochondrial proteins are upregulated in bat PaLung samples (109), with very few downregulated proteins (18), suggesting increased mitochondrial activity in PaLung cells.

To identify highly connected subnetworks that have strong expression changes, the set of 127 DE proteins was overlaid on a background of all known mitochondrial protein-protein interactions using STRING (with background obtained from MitoCarta and IMPI datasets). The results were clustered, and reactome pathway enrichment analysis of the resulting clusters showed enrichment for the following pathways: (1) TCA cycle and BCAA metabolism, (2) ETC, (3) fatty acid metabolism, and (4) protein import into mitochondria (*Figure 2C*).

We also performed GSEA using the abundances of the 1469 detected proteins as input against gene sets in the GO BP database. *Supplementary files 5–6* show the enriched gene sets in PaLung cells and WI-38 cells, respectively. *Supplementary file 5* shows that PaLung mitochondria were enriched for proteins in respiratory electron transport compared to WI-38 cells (*Figure 2D*). In particular, the gene set for Complex I biogenesis (*Figure 2E*) was significantly enriched in PaLung cells. For robustness, we also performed additional GSEA of the mitochondrial proteomics data under

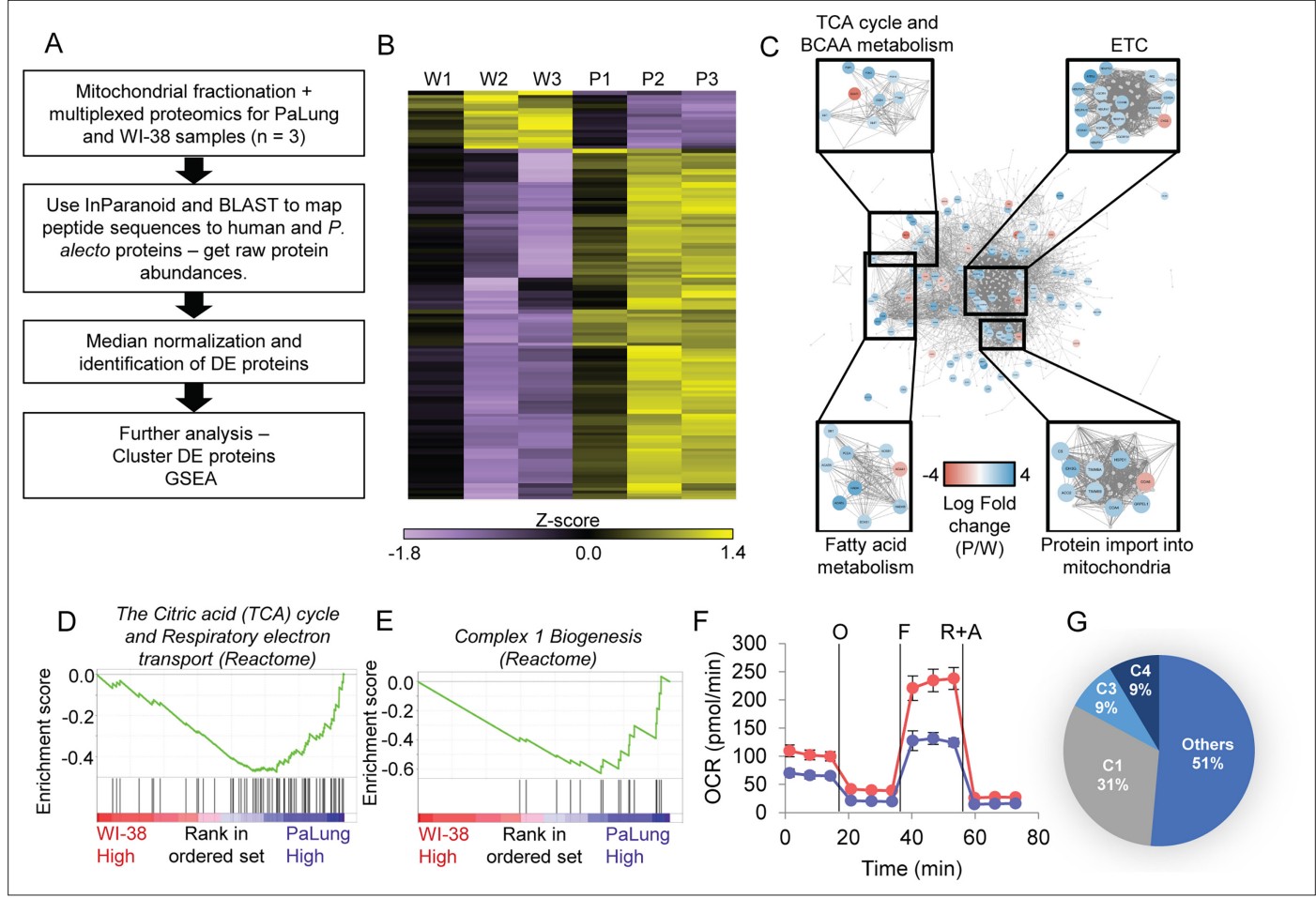

**Figure 2.** Proteomic data analysis of mitochondrial fractions of PaLung and WI-38 cells for differential expression, pathway enrichment, and electron transport chain (ETC) activity. (**A**) Workflow of bioinformatics analysis pipeline for proteomics data from PaLung (*P. alecto*) and WI-38 (*H. sapiens*) cells (*n*=3). (**B**) Heatmap showing the expression patterns for the 129 differentially expressed mitochondrial proteins in the three WI-38 samples (**W1–W3**) and the three PaLung samples (**P1–P3**). (**C**) Differentially expressed mitochondrial proteins (nodes colored by log fold change) are overlaid on a network of mitochondrial protein-protein interactions (obtained from STRING) (W=WI-38 cells, P=PaLung cells). The nodes are then clustered with respect to reactome-annotated pathways. (**D and E**) Gene set enrichment analysis (GSEA) identifies citric acid cycle, oxidative phosphorylation, and Complex I biogenesis as top metabolic pathways that are differentially regulated between PaLung and WI-38 cells. Shown here are the enrichment score plots for (**D**) citric acid cycle and oxidative phosphorylation and (**E**) Complex I biogenesis. (**F**) Oxygen consumption rate (OCR) measurement of PaLung cells (blue) and WI-38 cells (red) plotted as mean ± SD from n>15 independent experiments. O=oligomycin, F=FCCP, R+A = rotenone+antimycin A. (**G**) Pie chart showing that the proteomic upregulation of the ETC, implied by GSEA, is dominated by genes for subunits of Complex I. C1=Complex I; C3=Complex III; C4=Complex IV of the ETC.

The online version of this article includes the following figure supplement(s) for figure 2:

**Figure supplement 1.** Proteomic analysis of PaLung and WI-38 data.

different conditions, e.g., leaving out outlier samples P1, W1 (*Appendix 1—table 1*), or when using a mitochondria-specific gene set list instead of the full GO BP gene set list (*Appendix 2—table 1*). Results from these additional analyses agree with our primary GSEA that OxPhos and Complex I proteins are enriched in PaLung mitochondria compared to WI-38 mitochondria. Proteomics thus agrees with transcriptomics that OxPhos genes are more highly expressed in PaLung cells than WI-38 cells, however, there were no indications of hypoxia-related gene sets being differentially regulated in the proteomics dataset. Hence, we decided to pursue experimental studies of OxPhos rates in PaLung and WI-38 cells.

## PaLung cells have a lower OCR than WI-38 cells

To better characterize differences in OxPhos between PaLung and WI-38 cells, we monitored the oxygen consumption of these two cell lines using Seahorse XF Analyzer. Unexpectedly, the oxygen consumption rate (OCR) under basal conditions was lower in PaLung cells compared to WI-38 cells. Notably, bat cells have much lower maximal respiratory capacity compared to WI-38 cells, indicated by FCCP treatment (*Figure 2F*). This suggested that PaLung cells are able to carry out less OxPhos than WI-38 cells. While this observation agrees with earlier studies showing mild depolarization in the mitochondria of bats (*Vyssokikh et al., 2020*), it is in sharp contrast with our observation of increased OxPhos machinery in PaLung cells, using transcriptomics and proteomics. Taking a closer look at the omics results, we observed that the GSEA-flagged upregulation in OxPhos was driven mostly by the upregulation of Complex I subunits, for both the proteomic and transcriptomic data (*Figure 2G*, *Figure 1—figure supplement 1D*). This led us to hypothesize that in the basal state, PaLung cells might have partial decoupling of Complex I from the ETC, meaning that electrons emerging from Complex I might not proceed through the entirety of the ETC. This partial decoupling of Complex I might result in lower overall ATP synthesis, creating continued demand for ATP production from other sources. This would be consistent with our transcriptomic finding that both OxPhos-related and hypoxia-related genes were upregulated in PaLung cells. To build a self-consistent interpretation of these paradoxical omics and functional datasets, we proceeded to perform computational modeling of mitochondrial metabolism.

## Computational flux modeling suggests that Complex II of the ETC may run in reverse in PaLung cells

To understand the metabolic consequences of having higher Complex I activity but lower overall respiration, we turned to constraint-based flux modeling (*Figure 3—figure supplement 1*). We started with Mitocore, a published model of mitochondrial metabolism, that includes both core mitochondrial reactions and supporting cytoplasmic reactions from central carbon metabolism (glycolysis, pentose phosphate pathway, folate cycle, urea cycle, etc.) (*Smith et al., 2017*). Mitocore provides the set of metabolic reactions that can occur in a cell without specifying the activity, abundance, expression, or utilization of each element. To establish a model for each species, we took a species-specific subset of Mitocore reactions (called a *context-specific reconstruction*) based on the presence and absence of gene/protein expression in the transcriptomic/proteomic data for each species (see Materials and methods). This resulted in a PaLung model with 409 reactions and 324 metabolites (*Supplementary file 7*), and a WI-38 model with 437 reactions and 341 metabolites (*Supplementary file 8*) as shown in *Figure 3A*. We then performed uniform flux sampling (5000 samples) of the two models without imposing any constraints on either model (*Figure 3B*). Simulations resulted in a feasible flux distribution for each reaction in both the PaLung model and the WI-38 model, which can be depicted as a frequency histogram. In each chart in *Figure 3B and C*, the X-axis represents the metabolic flux value for the corresponding reaction, and Y-axis represents the frequency/probability of the reaction having the specific flux value. *Figure 3B* shows these histograms for the fluxes through Complex I–III in both the PaLung and WI-38 metabolic models, under unconstrained conditions (control simulation).

We next imposed the following two constraints, which are hypothetical differences between bat and human cells, suggested by the above transcriptomic/proteomic and oxygen consumption measurements. The first constraint is that the PaLung model must have greater activity of Complex I than the WI-38 model. This constraint was inspired by the transcriptomic/proteomic data (*Figures 1C and 2F*) and by inferring that the flux through Complex I in PaLung cells would be greater than the flux through Complex I in WI-38 cells. (see 'Constraints for flux sampling' in Materials and methods). The second constraint, based on our oxygen consumption data, is that the WI-38 model must have higher oxygen intake into the mitochondria than the PaLung model. When the PaLung and WI-38 models were simulated under these two constraints, the flux histograms for PaLung cells had shifted to very low or negative flux values for Complex II (*Supplementary file 9*). Complex II is also called succinate dehydrogenase (SDH) and is part of both the TCA cycle and the ETC. SDH generally catalyzes the conversion of succinate into fumarate, accompanied by a reduction of the endogenous Quinone pool. However, SDH has also been documented to catalyze the reverse reaction, converting fumarate to succinate, although this is unconventional. Such an unconventional ETC paradigm where Complex I proceeds forward but Complex II proceeds in reverse (toward the accumulation of succinate) has been

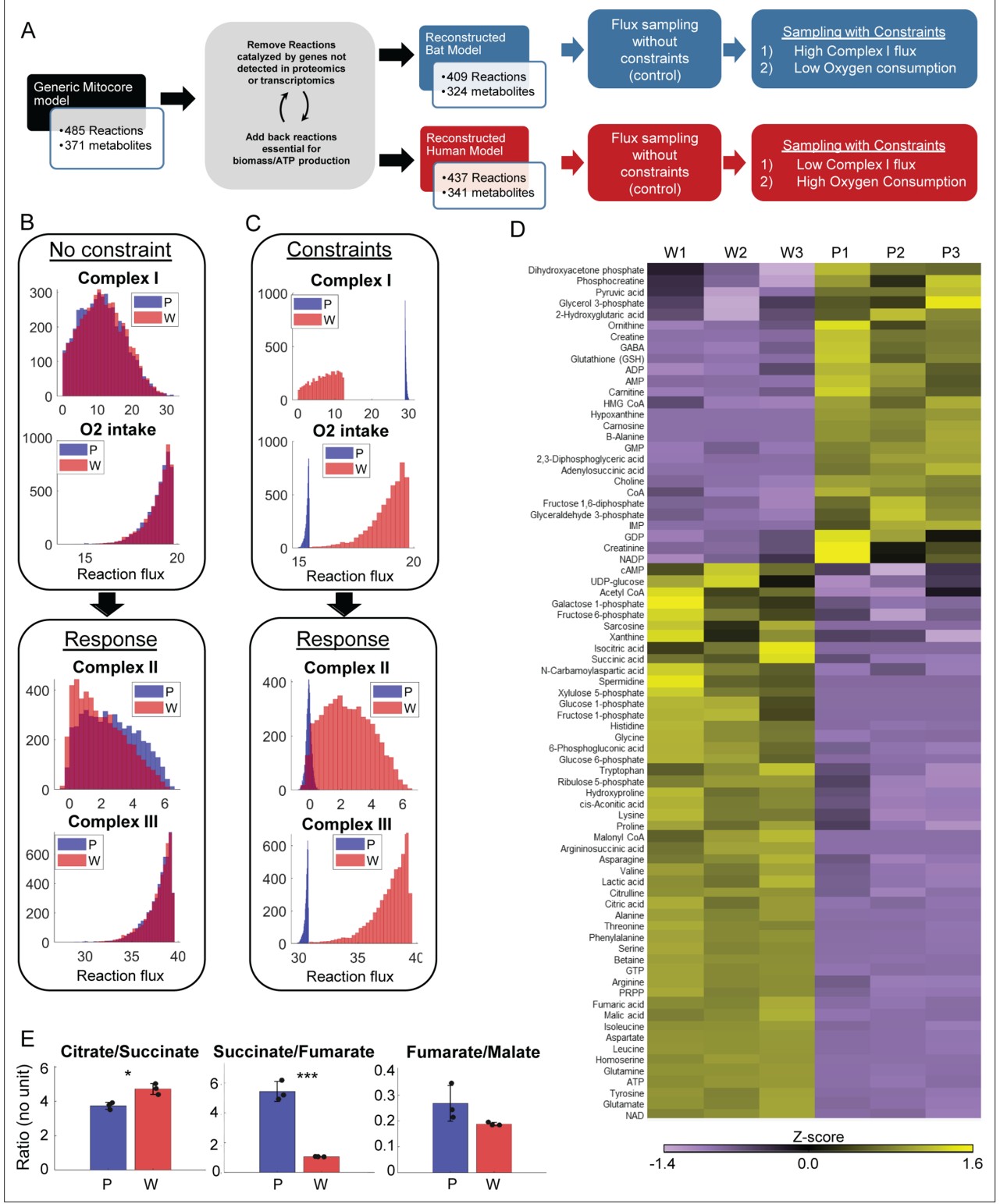

**Figure 3.** Metabolomic data and model-based analysis of mitochondrial metabolism in PaLung and WI-38 cells. (**A**) Schematic showing the metabolic modeling pipeline. We begin with the context-specific reconstruction of a metabolic model – the process where a generic mitochondrial model (**Smith et al., 2017**) is tailored specifically to PaLung and WI-38 cells using proteomic and transcriptomic expression patterns. The individual metabolic models are then simulated using constraint-based flux sampling methods to give a distribution of possible fluxes for each reaction in the model. Comparing these flux distributions between the two models allows the detection of metabolic reactions that are likely to be differentially regulated in response to user-imposed constraints on metabolism. Simulations are performed on both the PaLung and WI-38 models under no constraints or with constraints

*Figure 3 continued on next page*

*Figure 3 continued*

on Complex I and mitochondrial $O_2$ intake. (**B**) Sample histograms showing the feasible flux distributions for electron transport chain (ETC) reactions in the unconstrained PaLung (**P**) and WI-38 (**W**) metabolic models. (**C**) Flux distributions of ETC reactions in PaLung (**P**) and WI-38 (**W**) cells when PaLung cells are constrained to have higher flux through Complex I of ETC but lower oxygen intake in the mitochondria. (**D**) Heatmap showing differentially regulated metabolites from central carbon metabolism in the three WI-38 (**W1–W3**) and the three PaLung (**P1–P3**) samples. (**E**) Intra-sample ratios of metabolites from the TCA cycle in either PaLung (**P**) or WI-38 (**W**) cells, plotted as mean ± SD from three independent experiments. * and *** represent p-value≤0.05 or ≤0.001 respectively (unpaired Student's two-sided t-test with Benjamini-Hochberg correction for multiple hypothesis testing).

The online version of this article includes the following figure supplement(s) for figure 3:

**Figure supplement 1.** Schematic of the workflow for metabolic flux modeling.

**Figure supplement 2.** Analysis of the metabolomics data from PaLung and WI-38 cells.

observed in other systems, e.g., in murine retinal tissues (*Bisbach et al., 2020*) and human hearts (*Chouchani et al., 2014*), under ischemic or hypoxic conditions. In these cases, the resulting accumulation of succinate was found to be useful in fueling other cells in the tissue or in avoiding reperfusion injury when ischemic conditions were abruptly removed. During conventional ETC, both Complex I and Complex II operate in parallel and produce electrons that are shuttled downstream to Complex III, Complex IV, and ATP synthase. However, prior work (*Bisbach et al., 2020*; *Chouchani et al., 2014*) has documented an alternative in which the electrons obtained from Complex I can be consumed by Complex II operating in reverse, rather than traversing the rest of the ETC. Accordingly, the low or negative flux values for Complex II in our PaLung simulations indicate that the electrons obtained from Complex I may accumulate at Complex II or potentially even get consumed by Complex II operating in reverse (bypassing the rest of the ETC) in PaLung cells. To further interrogate central carbon metabolism in PaLung cells and to validate if our predictions of unconventional SDH activity might be borne out by metabolic measurements, we undertook targeted metabolomics of small organic compounds for both PaLung and WI-38 cells.

## Metabolomics supports the possibility of low/reverse Complex II activity in PaLung cells

Mass spectrometry-based targeted metabolomics was performed on both PaLung and WI-38 cell lines, resulting in the absolute quantification of 116 metabolites from central carbon metabolism (*Figure 3D*, *Figure 3—figure supplement 2*, *Supplementary file 10*). Since we were comparing different species, we looked at relative ratios of metabolites within each species. As predicted by our hypothesis, metabolomics showed that the ratio of succinate-to-fumarate was much higher in PaLung cells (5.44±0.67) compared to WI-38 cells (1.07±0.012) (p<0.001), consistent with succinate accumulation in PaLung cells (*Figure 3E*). In contrast, the ratio of other serial TCA metabolites, e.g., citrate/succinate or fumarate/malate, showed only mild or insignificant differences between PaLung and WI-38 cells. We interpreted this as an indication that the TCA cycle acts in a truncated manner in PaLung cells, with SDH operating in reverse to support succinate accumulation. Interestingly, previous studies have documented such SDH phenomena (with the low or reverse activity of Complex II) during ischemic/hypoxic states that had low metabolic rates and high levels of AMP (*Chouchani et al., 2014*; *Bisbach et al., 2020*). This led us to wonder if PaLung cell metabolism might resemble an ischemic-like state despite oxygen and nutrient availability, in which case we would expect to see low metabolic rates and high AMP levels in PaLung cells.

## PaLung cells exhibit basal metabolism that resembles an ischemic-like state and can tolerate glucose deprivation better than WI-38 cells

To further understand the consequences of the truncated TCA cycle and its implications on an ischemic-like state, we looked at the abundances of other metabolites in metabolomics datasets. Looking at metabolites from glycolysis, pentose phosphate pathway, TCA cycle, and the levels of amino acids, we found that most metabolites were at much lower levels in PaLung cells compared to WI-38 cells, except for a small subset of metabolites in glycolysis (*Figure 4A–C*). This suggested an overall slower metabolic turnover in PaLung cells. PaLung cells also had a much higher proportion of AMP (*Figure 4D*), which would be expected in an ischemic setting.

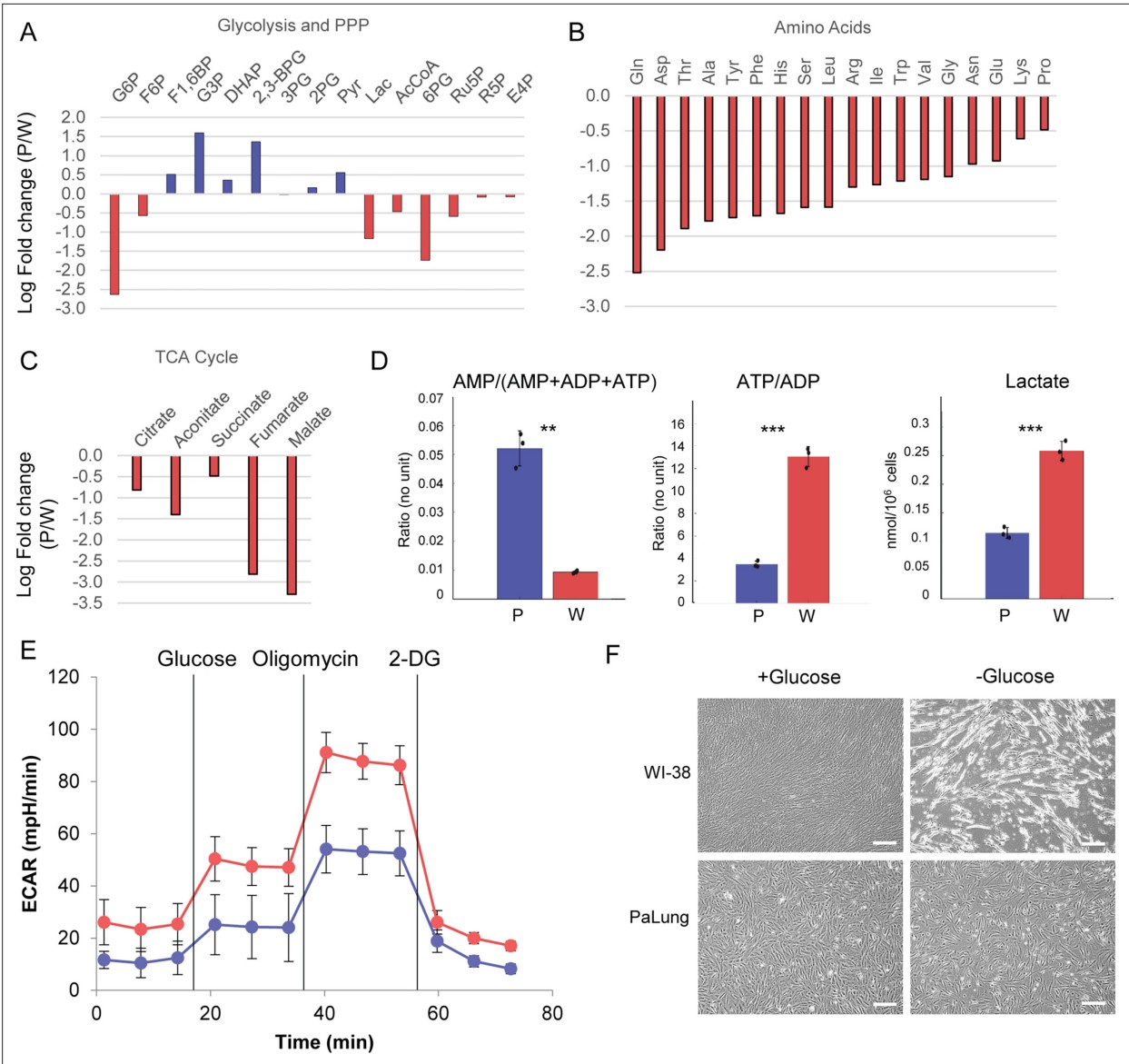

**Figure 4.** PaLung cells show basal metabolism that resembles an ischemic-like state. (**A**) Fold changes (mean PaLung/mean WI-38) of metabolite abundances, obtained through targeted metabolomics profiling of central carbon metabolites in PaLung and WI-38 cells. (**B and C**) Amino acid and metabolite changes in TCA cycle (PaLung/WI-38) in PaLung cells and WI-38 cells. (**D**) Bar plots of AMP/(AMP+ADP+ATP), ATP/ADP ratio, and lactate amounts in PaLung (**P**) and WI-38 (**W**) cells plotted as mean ± SD from three independent experiments. ** and *** represent p-value≤0.01 or ≤0.001 respectively (unpaired Student's two-sided t-test with Benjamini-Hochberg correction for multiple hypothesis testing). (**E**) Extracellular acidification rate (ECAR) measurement of PaLung (blue) and WI-38 (red) cells plotted as mean ± SD from n>15 independent experiments (2-DG=2-deoxy-D-glucose). (**F**) Phase contrast images of PaLung and WI-38 cells with or without glucose deprivation for 96 hr. Scale bar, 100 μm.

Furthermore, our metabolomics dataset revealed that PaLung cells had a threefold lower ATP/ADP ratio compared to WI-38 cells (p<0.001). ATP/ADP ratio is a golden standard for the measurement of cellular energy status, and our results suggest lower metabolism in PaLung cells. Another corroborating measurement for energy status is adenylate energy charge, with lower values indicating slower metabolism (*Atkinson and Walton, 1967*). We found that PaLung cells had lower adenylate energy charge compared to WI-38 cells (p<0.001) (*Supplementary file 10*). From these parameters, we infer that PaLung cells have lower ATP synthesis, consistent with our earlier observations of lower OCR in PaLung cells (*Figure 2F*). To check if slower metabolism would also translate to slower glycolysis in PaLung cells, we first checked our transcriptomics and metabolomics data. Both omics datasets showed mixed signals along the glycolysis pathway (with partial upregulation and partial

downregulation) (*Figure 4A* and *Figure 1—figure supplement 2*), yielding no predictions for glycolysis utilization. However, metabolomics showed higher levels of intracellular lactate in WI-38 cells than in PaLung cells, suggesting that PaLung cells could have lower glycolytic flux than WI-38 cells. Finally, we assessed glycolysis levels using a Seahorse XF Analyzer to quantify the extracellular acidification rate (ECAR). This revealed that PaLung cells had lower ECAR compared to WI-38 cells (*Figure 4E*). From these observations of low ECAR, low lactate, and low ATP/ADP ratio, combined with the earlier findings of low OCR, we conclude that PaLung cells have less energy production and a lower level of basal metabolic activity than WI-38 cells.

Reduced OxPhos and glycolysis in PaLung cells somewhat resemble an ischemic-like state with inadequate oxygen and glucose supply. Therefore, PaLung cells may be resistant to metabolic stress. To test this, we subjected PaLung and WI-38 cells to glucose deprivation for 96 hr. Interestingly, PaLung cells displayed higher viability than WI-38 cells after glucose deprivation (*Figure 4F*), despite starting with lower stores of internal energy (e.g. glucose-6-phosphate, acetyl-CoA, and lower adenylate charge). These data suggest that PaLung cells do have a better ability to tolerate metabolic stress compared with WI-38 cells, consistent with low basal metabolic activity.

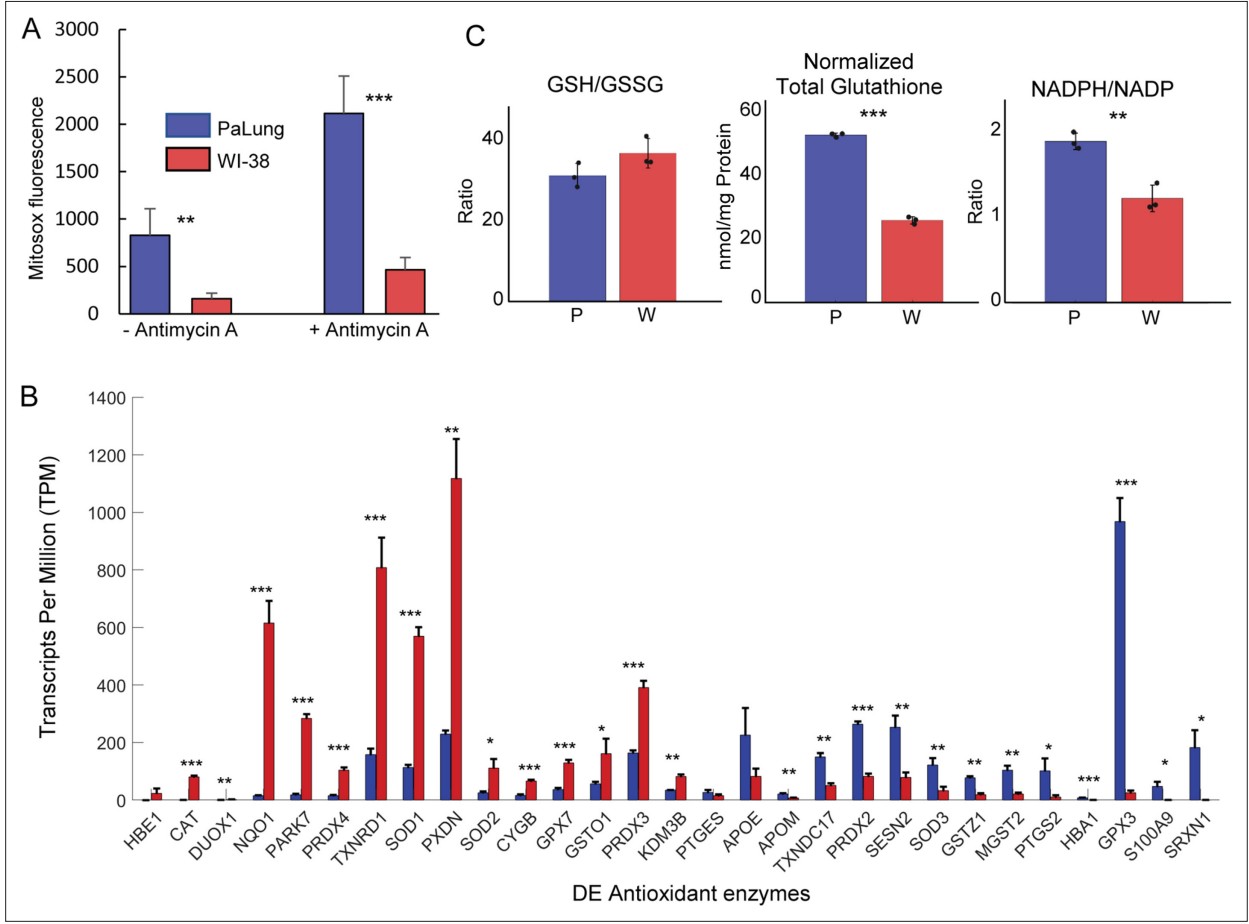

**Figure 5.** Reactive oxygen species (ROS) and antioxidant system measurements in PaLung and WI-38 cells. (**A**) MitoSOX measurement of PaLung and WI-38 cells with or without antimycin A treatment for 1 hr. Antimycin A is an electron transport chain (ETC) inhibitor known to induce superoxide generation. (**B**) Bar charts showing the expression levels of antioxidant genes that passed our differential expression thresholds (as transcripts per million [TPM]) in PaLung (blue) and WI-38 (red) cells. Genes have been sorted in increasing order of P/W fold change. (**C**) Bar plots show the ratio of reduced to oxidized glutathione (GSH/GSSG), total glutathione normalized to protein content (GSH+GSSG), and the ratio of NADPH/NADP in PaLung (**P**) and WI-38 (**W**) cells. For all panels, bars are the mean ± SD from three independent experiments (*n*=3). *, **, or *** represents p-value<0.05, ≤0.01, or ≤0.001 respectively (unpaired Student's two-sided t-test with Benjamini-Hochberg correction for multiple hypothesis testing).

The online version of this article includes the following figure supplement(s) for figure 5:

**Figure supplement 1.** Increased phosphorylation of NAD cofactors in PaLung cells.

## PaLung cells have higher ROS compared to WI-38 cells but lower expression of many antioxidant genes

Previous studies of ischemic metabolism, in which SDH functions in reverse, have showed that high levels of ROS were generated as a result. To test if the same would be true in PaLung cells, we performed the MitoSOX assay to measure superoxide levels in the mitochondria of PaLung cells and WI-38 cells. Indeed, PaLung cells showed higher mitochondrial superoxide levels compared to WI-38 cells, both under basal conditions and when ETC was inhibited using antimycin A (*Figure 5A*). Searching our transcriptomic data for genes involved in the redox control, we observed that there were significant differences in the expression of redox control genes in PaLung and WI-38 cells (*Figure 5B*). Notably, SOD1 and SOD2, key enzymes that convert mitochondrial superoxide to the more toxic intracellular ROS, are less expressed in PaLung cells than in WI-38 cells, consistent with higher mitochondrial superoxide levels in PaLung cells. Interestingly, glutathione peroxidase 3 (*GPX3*), a well-known antioxidant enzyme that reduces hydrogen peroxide or organic hydroperoxides using glutathione, was found to be highly upregulated in PaLung cells compared to WI-38 cells. This led us to test levels of glutathione, a non-enzyme antioxidant for ROS detoxification, in the two cell lines.

## PaLung cells have a robust glutathione NADPH system to counter intracellular ROS

To test if the glutathione system may help PaLung cells tolerate intracellular ROS, we measured intracellular glutathione and NADP(H) levels in both PaLung and WI-38 cells. The ratio of reduced to oxidized glutathione (GSH/GSSG ratio) was not significantly different between the two cell lines (WI-38 cells: 36.45±3.6 vs PaLung cells: 30.94±2.97). However, PaLung cells had a twofold higher concentration of total intracellular glutathione, compared to WI-38 cells (*Figure 5C*). We also found that PaLung cells had a higher NADPH/NADP ratio (1.5-fold) compared to WI-38 cells. Overall, the metabolomics data indicated that PaLung cells had a nearly 2.5-fold higher ratio of NADP/NAD, i.e., a higher resting concentration of phosphorylated to unphosphorylated NAD (*Figure 5—figure supplement 1A*). This was also supported by transcriptomics which showed an upregulation of NADK (NAD kinase) and supporting enzymes required to synthesize and phosphorylate NAD (*Figure 5—figure supplement 1B*). Taken together these results suggest that PaLung cells maintain a higher standing pool of glutathione and a higher NADPH concentration to counter ROS generated due to ischemic-like metabolism.

## PaLung cells are resistant to ferroptosis

Many earlier studies have shown that ischemic conditions can induce cell death via ferroptosis, which also depends on accumulated ROS and low glutathione levels (*Chen et al., 2021*). We wondered if apart from combatting ROS, the high glutathione levels might also help PaLung cells avoid ischemia-induced ferroptosis. Recent studies have reported that ischemia-induced ferroptosis causes tissue damage and that inhibition of ferroptosis attenuates ischemia-induced cell death (*Xie et al., 2019*; *Liao et al., 2021*). Ferroptosis is a non-apoptotic, programmed form of cell death, which is iron-dependent and occurs via glutathione depletion-induced lipid peroxidation (*Ursini and Maiorino, 2020*). Given that PaLung cells showed upregulation of glutathione/NADPH antioxidant system and genes related to hypoxia response (*Figure 1D*) and ischemic metabolism, we tested whether PaLung cells can better tolerate ferroptosis-inducing conditions. Ferroptosis was induced by erastin, an SLC7A11/xCT inhibitor. While WI-38 cells showed high sensitivity to erastin-induced ferroptosis that were prevented by a ferroptosis inhibitor ferrostatin-1, PaLung cells were resistant to erastin-induced ferroptosis (*Figure 6A and B*). Similarly, PaLung cells were more resistant to cystine deprivation-induced ferroptosis, compared to WI-38 cells and had almost sevenfold lower cell death than WI-38 cells (3.5% cell death in PaLung cells compared to 24.4% cell death in WI-38 cells upon cystine deprivation) (*Figure 6C and D*). These results suggest that PaLung cells are more resistant to ferroptosis compared to WI-38 cells.

To better understand the link between glutathione concentrations and ferroptosis resistance in PaLung cells, we measured the amounts of glutathione in WI-38 and PaLung cells under cystine deprivation. Both WI-38 and PaLung cells showed decreased GSH/GSSG ratio under cystine deprivation, which indicates both cell lines are under oxidative stress (*Figure 6E*). Cystine deprivation decreased total glutathione levels in WI-38 cells by 94.5%, nearly depleting glutathione reserves. In contrast,

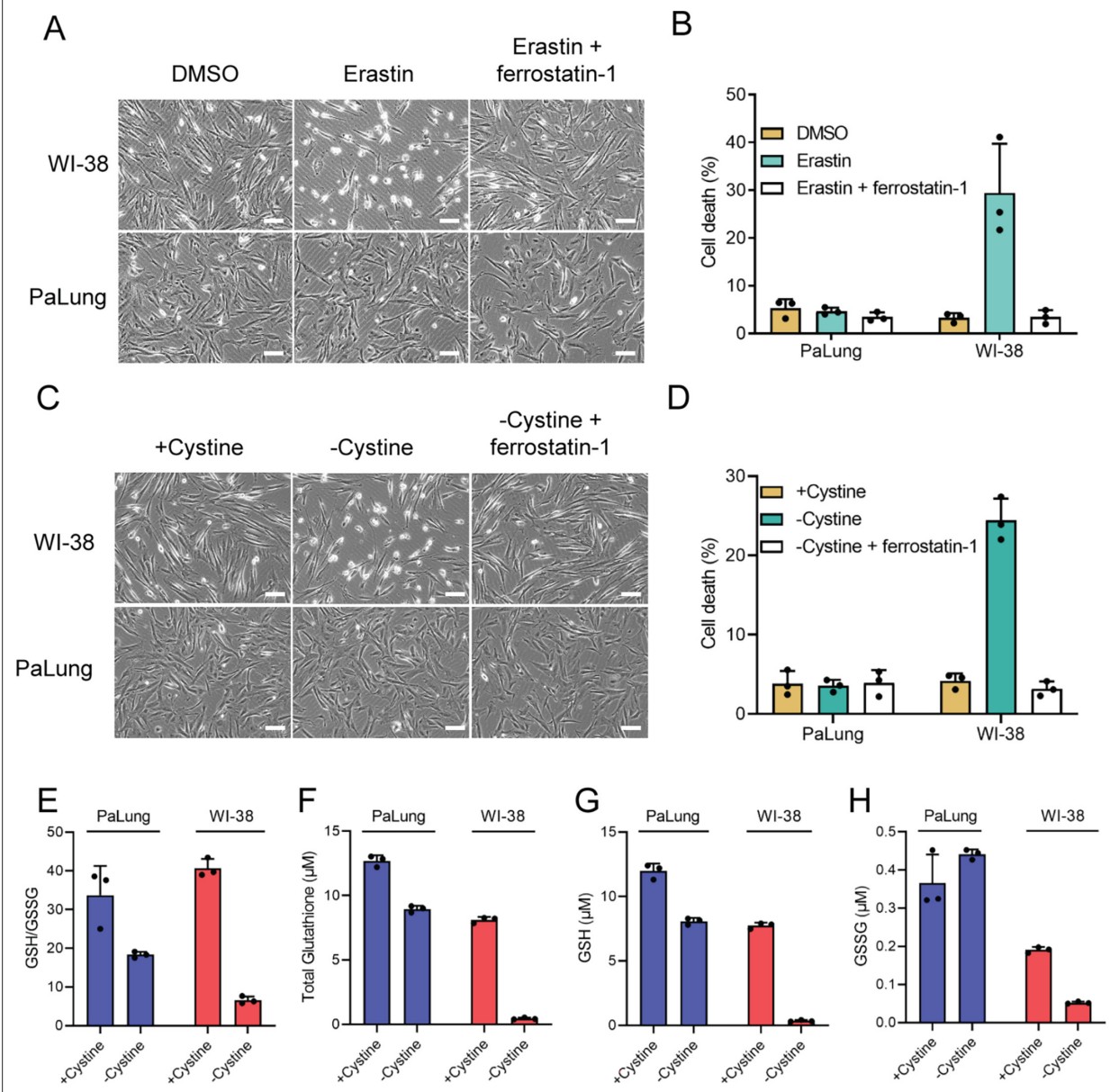

**Figure 6.** PaLung cells display high resistance to ferroptosis. (**A and B**) WI-38 or PaLung cells were treated with 2.5 µM erastin and/or 1 µM ferrostatin-1. Representative images were taken at 6 hr using phase contrast microscopy (**A**). Propidium iodide (PI) exclusion assay was performed 24 hr after erastin treatment (**B**). (**C and D**) WI-38 or PaLung cells were cultured in media with or without cystine. Ferrostatin-1 (1 µM) was treated simultaneously. Representative images were taken at 8 hr using phase contrast microscopy (**C**). PI exclusion assay was performed at 24 hr after cystine deprivation (**D**). (**E–H**) WI-38 or PaLung cells were cultured in media with or without cystine for 6 hr. Intracellular glutathione levels were measured. Reduced glutathione (GSH)/oxidized glutathione (GSSG) ratio (**E**), total glutathione (the sum of GSH and GSSG) (**F**), GSH (**G**), and GSSG (**H**) levels were measured. Scale bars, 50 µm. The mean ± SD of three independent experiments is shown.

cystine deprivation in PaLung cells resulted in only a modest 29.5% decrease, and PaLung cells still maintained a total glutathione level higher than non-cystine-deprived WI-38 cells. The higher levels of total glutathione, GSH, and GSSG under cystine deprivation might explain the high resistance of PaLung cells under ferroptosis-inducing conditions (*Figure 6F, G, and H*).

## Discussion

In this study, we integrated high-throughput omics and computational metabolic modeling to perform a novel comparison of central metabolism between cell lines from two mammalian species.

Specifically, we have identified core differences in mitochondrial metabolism between primary lung fibroblasts of the black flying fox fruit bat *P. alecto* (PaLung) and the primary human lung fibroblast cell line WI-38. Although data are still limited to these two cell lines, this is the first comprehensive analysis combining proteomics, transcriptomics, metabolomics, and constraint-based flux modeling between human and bat cells. Our analysis suggests that PaLung cells exhibit basal metabolism that resembles an ischemic-like state and that this state may be linked to low or reverse activity of Complex II of the ETC (also called SDH). Compared to WI-38 cells, PaLung cells also show a higher tolerance to cellular stresses such as nutrient deprivation and ischemia/ROS-driven cell death via ferroptosis.

Hypoxia response (including glycolysis) and OxPhos compensate for each other by gene expression, dependent on different levels of oxygen availability. However, our analyses (GSEA from whole-cell transcriptomics) indicated that genes related to glycolysis and OxPhos were simultaneously upregulated in PaLung cells (bat) compared to WI-38 cells (human). Although multiple glycolytic genes were upregulated in PaLung cells (*Figure 1—figure supplement 2*), lactate production (inferred via both LDH expression levels and ECAR measurement) in PaLung cells was not higher than in WI-38 cells. This raises an important caveat about using ECAR to infer glycolytic flux, because ECAR depends only on the amount of lactate produced, and glycolytic pyruvate that enters the TCA cycle may remain invisible to ECAR measurements.

Our observation about the upregulation of OxPhos in PaLung cells via GSEA (transcriptomics and proteomics) also raises an important point of concern, namely that high signals of a certain pathway in GSEA can be driven by a narrower subset of genes. For instance, in both our transcriptomic and proteomic GSEA results, the ETC genes upregulated in the PaLung samples did follow the frequency distribution expected when taking into account the total subunit counts of each ETC complex (CI: 44, CII: 4, CIII: 10, and CIV: 19 subunits). However, the sheer absolute number of Complex I genes upregulated was enough for GSEA to declare the entire OxPhos gene set as upregulated in PaLung cells. While this does not say if the other ETC complexes were upregulated in PaLung, it shows the outsized effect Complex I genes have on the entire OxPhos pathway, simply because of overwhelming numbers. Hence, while interpreting GSEA results, it is necessary to survey the constituent genes that are responsible for the predicted up/downregulation of the overall pathway. Indeed, we found that the upregulation of OxPhos pathway in PaLung cells in GSEA was due to high gene expression of the Complex I components in the ETC. However, our experimental analysis revealed a puzzling result: despite high Complex I expression, oxygen consumption was low in PaLung cells.

To interpret the metabolic implications of heightened Complex I and lower oxygen consumption, we turned to constrained-based metabolic flux sampling. Flux sampling is a technique that can simulate possible states of a metabolic network. Compared to higher resolution methods such as isotope-labeled fluxomics, flux sampling is a coarse-grained qualitative approach to study metabolic flux. However, it can be used as a tool to generate testable hypotheses in exploratory studies such as ours. Indeed, metabolomics was able to verify our simulation predictions of succinate accumulation in PaLung cells, a phenomenon that could result from low/reverse activity (fumarate to succinate) of Complex II of ETC (SDH). The succinate-to-fumarate ratio in well-oxygenated PaLung cells was found to be similar to those found in ischemic states of human cells from multiple tissues (*Chouchani et al., 2014*). From this, we infer that the TCA cycle in PaLung cells does something different than the TCA cycle in human WI-38 cells, which is also corroborated by our observation that PaLung cells have a low metabolic rate. We also confirmed using cell culture assays that PaLung cells have a much longer survival during glucose deprivation than WI-38 cells.

We also found that PaLung cells had higher loads of mitochondrial ROS, consistent with having the lower expression of *SOD1* and *SOD2*. On the other hand, PaLung cells highly express glutathione peroxidase 3 (*GPX3*), contain twofold higher levels of glutathione, and maintain a higher NADPH/NADP ratio compared to WI-38 cells. These data suggest that PaLung cells maintain a standing pool of NADPH and glutathione that can act against high levels of mitochondrial ROS. Therefore, the previously published generalization that bats have lower free radical production than other mammals (*Brunet-Rossinni, 2004*; *Brown et al., 2009*) may require moderation. Future work can address whether PaLung cells have a higher threshold for homeostatic ROS and whether ROS have both beneficial and detrimental effects in bats, as has been observed in other organisms (*Clément and Pervaiz, 2001*; *Shields et al., 2021*).

The key novelty of our results is the suggestion of a basal ischemic-like state in PaLung cells accompanied by higher ROS production and higher glutathione and NADPH levels. Multiple studies have shown strong links between ischemia, ROS, and ferroptosis (an iron-dependent, non-apoptotic form of cell death). Despite having ischemic-like metabolism and higher ROS levels, we observed that PaLung cells were more resistant to ferroptosis than WI-38 cells. Ferroptosis occurs via ROS-induced lipid peroxidation following ischemia and is conventionally blocked via the action of glutathione peroxidase 4 (*GPX4*). While *GPX4* expression levels were low in PaLung cells, *GPX3* (a commonly secreted form of glutathione peroxidase) was highly expressed. Future studies are needed to determine the function of GPX3 in the stress response and whether it contributes to the ferroptosis resistance of PaLung cells.

Our identification of a metabolic state in bats that resembles ischemia is also tied to conventional wisdom about bat flight. During flight, the metabolic rate of bats can go 2.5–3 times higher than that of other mammals, consuming approx. 1200 calories per hour, resulting in an immense drain on stored energy reserves, and the potential depletion of up to 50% of stored energy in fructivore bats (*Thomas, 1975*; *Voigt and Speakman, 2007*; *Kelm et al., 2011*). Thus, a basal state with a low metabolic rate may serve to conserve energy during non-flying periods. In addition, the accumulated succinate during this ischemic-like state can also be transported into the bloodstream and serve as a metabolic stimulant for other fast-metabolizing cell types as needed, similar to examples of succinate shuttling between two cell types in retinal tissues (*Bisbach et al., 2020*). Low metabolic rates have also been linked to longevity by many studies, which demonstrate that calorie restriction contributes to an extended lifespan across various species. In our experiments, *P. alecto* also showed higher expression of genes involved in NAD synthesis and its phosphorylation pathways, in agreement with other studies that have linked higher flux through the NAD biosynthesis pathway with slower aging (*Chini et al., 2017*).

Given that we compare two vastly different species, our work is subject to some technical limitations due to assumptions of comparability. Specifically, (1) we perform GSEA using human-derived GO terms and gene sets, (2) we map proteomic peptides between human and *P. alecto* proteins, and (3) we use human-derived metabolic mitochondrial models for computational flux sampling. However, such assumptions are unavoidable and currently serve as the best recourse, until further research into bats can provide better bioinformatic tools more suited to bats. In addition, reference genomes for less studied organisms like bats may not be as well annotated as other organisms. Since we perform transcriptomic analyses only using genes identified in both species, observed downregulation of a pathway in PaLung cells might be because the constituent genes were not annotated as well, and not because they weren't expressed as highly. For exactly this reason, we have focused on pathways that were upregulated in bats, and we made no conclusion about pathways downregulated in bats.

A limitation of our study is that we performed multi-omics comparisons between individual cell lines. Extending the current study to primary tissues or cell lines from other species of bats might provide further information about generalized metabolic differences between bats and humans. The differences we observe in our study could also be cell-type dependent. For example, our choice of fibroblast cell lines may explain why we observed PaLung cells to have lower oxygen consumption than WI-38 cells, even under uncoupled conditions. Future work can study muscle cells in the same species to obtain broader metabolic insights. The current study can be considered a starting point to both generate hypotheses for future work and to establish analysis pipelines for inter-species comparisons of metabolism, using multi-omics data and computational flux modeling simulations.

In summary, using multi-omics datasets (transcriptomic, proteomic, metabolomic) and computational modeling, we have compared the basal metabolic states of a human cell line and a *P. alecto* cell line. Our work points toward important differences in ETC regulation (potential low/reverse Complex II activity) and antioxidant response (higher glutathione and NADPH) that could contribute to ischemia/redox management and ferroptosis resistance in *P. alecto*. Future research can extend the idea further to identify if such regulation is a pan-bat feature and if such metabolic fingerprints might also contribute to cancer resistance and increased longevity in bats.

## Materials and methods

### Cell cultures and reagents

WI-38 cells (RRID: CVCL_0579) from *H. sapiens* (RRID: NCBITaxon_9606) were purchased from Coriell Institute. PaLung, a lung-derived cell line from *P. alecto* (RRID: NCBITa xon_9402), was established as previously described (*Koh et al., 2019*; *Crameri et al., 2009*). In brief, PaLung cells were derived from the lung tissue of a single female *P. alecto* (*n*=1) and were established as a primary cell line through a process that combined trypsinization and physical disruption. This was followed by culturing in Dulbecco's modified Eagle's medium (DMEM)/F12-Hams (Sigma) medium, supplemented with 15% bovine calf serum (Hyclone), 100 units/ml penicillin, 100 µg/ml streptomycin, and 50 µg/ml gentamycin (Sigma) in 5% $CO_2$-humidified atmosphere at 37°C. Upon their establishment, all the cell lines were cultured in high-glucose DMEM (#11965, Gibco, Life Technologies) supplemented with 10% FBS (HyClone, GE Healthcare Life Science), penicillin (100 units/ml), and streptomycin (100 mM/ml; Gibco, Life Technologies) in 5% $CO_2$-humidified atmosphere at 37°C. WI-38 cells were authenticated by Coriell Institute. The PaLung cell line is a novel primary cell line derived from bats and was not authenticated since no standard workflow has been established for authentication of bat-derived cells. WI-38 and PaLung cell lines tested negative for mycoplasma contamination. For glucose deprivation, cells were washed with phosphate-buffered saline (PBS) three times and cultured in glucose-free DMEM (#10966, Gibco, Life Technologies) with 10% dialyzed FBS. For cystine deprivation, cells were washed with PBS and cultured in cystine, methionine, glutamine-free DMEM (#21013024, Gibco, Life Technologies) supplemented with 0.2 mM L-methionine, 4 mM L-glutamine, and 10% dialyzed FBS. L-glutamine was purchased from Invitrogen. L-methionine and L-cystine were kindly provided by Dr. Jean-Paul Kovalik (Duke-NUS Medical School, Singapore). Oligomycin, FCCP, rotenone, antimycin A, 2-deoxyglucose, and erastin were purchased from Sigma-Aldrich. Ferrostatin-1 was purchased from Med Chem Express.

### PI exclusion assay

Cells were stained with PI to determine the percentage of cell death. Media containing floating cells were collected, combined with trypsinized cells, and centrifuged. The cell pellet was washed once with PBS. After centrifugation, cells were resuspended and stained with PI (10 µg/ml) for 10 min at room temperature. Data were collected with MACSQuant analyzer (Miltenyi Biotec). Quantification and analysis of the data were done with FlowJo software (RRID: SCR_008520).

### Transcriptomics (sample preparation and initial bioinformatics)

Three independent sets of RNA were collected from WI-38 and PaLung cells at different passages (*n*=3). In brief, cells were subcultured at a 1:2 ratio and collected at passage numbers 26 (W1), 30 (W2), and 35 (W3) for WI-38 cells, and 5 (P1), 6 (P2), and 7 (P3) for PaLung cells. RNA was extracted from WI-38 and PaLung cells using RNeasy Plus Mini Kit (QIAGEN). 1000 ng of total RNA from each sample was used to generate RNAseq libraries using TruSeq Stranded Total RNA Library Prep Gold according to the manufacturer's instructions (Illumina). Library fragment size was determined using DNA1000 Assay on the Agilent Bioanalyzer (Agilent Technologies). 2×150 PE sequencing was subsequently performed on the libraries using HiSeq3000 equipment (Illumina). The resulting reads were cleaned/trimmed and demultiplexed, followed by mapping to either the PA genome (annotation 102, 2018) or the hg19 genome using cufflinks/Tophat.

### Transcriptomics, DEseq, GeTMM, and GSEA

For DE analysis, the raw read counts were input to the R package DESeq2 (*Love et al., 2014*). Genes with counts per million < 1 in more than three out of six samples (three from PaLung and three from WI-38) were discarded from downstream analysis. To account for differences in transcript length between the two species, the individual transcript lengths were supplied as an additional normalization factor to DESeq2. Since the inter-species analysis of RNAseq data does not have conventional workflows, we repeated DE gene identification using an alternative workflow with GeTMM (*Smid et al., 2018*). This method uses RPKM as input and hence accounts automatically for differences in transcript lengths. This workflow was implemented using the R package EdgeR (*Robinson et al., 2010*). For GSEA, we used the GSEApreranked module in GSEA version 4.1.0 (*Subramanian et al., 2005*), using the DESeq2 results as input. For ranking genes, we used the π-value metric $[LFC * (-log_{10}(pvalue))]$

(LFC = log$_2$ fold change) (*Xiao et al., 2014*). GSEA was run against the complete gene set list that includes GO BP with pathways from databases such as Reactome and KEGG (containing 18,356 gene sets), downloaded from https://download.baderlab.org/EM_Genesets.

## Mitochondria isolation

Mitochondria were isolated from WI-38 and PaLung cells using mitochondrial isolation kit from Miltenyi Biotech as described in the manufacturer's protocols. In brief, $1\times10^7$ cells from WI-38 and PaLung cells were lysed in the ice-cold hypotonic lysis buffer for 60 s followed by mechanical disruption of cell membrane using a mini homogenizer pestle gun for 60 s. The suspension was centrifuged at 700 × *g* for 5 min at 4°C and supernatant was collected. The supernatant containing the mitochondria was incubated with the TOM-22 antibody-conjugated with MACS magnetic beads (Miltenyi Biotech) and pulled down using the magnetic columns. The mitochondria fractions were eluted with 0.1 M glycine pH 3.5, neutralized with Tris-HCl pH 7.5, and stored at 80°C until the time of proteomics profiling.

## Proteomics profiling and analysis

The mitochondrial fractions were lysed in 8 M urea pH 8.5 and incubated with 20 mM tris(2-carboxyethyl)phosphine hydrochloride for 20 min at 25°C followed by alkylation with 55 mM chloroacetamide at 25°C for 30 min. Proteins were digested with trypsin overnight at 25°C. Digested peptides were acidified with 1% trifluoroacetic acid, desalted on C18 plates Oasis (Waters), and labeled with TMT sixplex reagent (Thermo Scientific) according to the manufacturer's protocol. Labeled samples were further fractionated with high pH reverse phase using spin columns packed in-house, and five fractions were collected: 10%, 17.5%, 25%, 30%, and 50%. The fractions were separated on a 50 cm × 75 µm Easy-Spray column using Easy-nLC system coupled with an Orbitrap Fusion Tribrid mass spectrometer (Thermo Scientific). The LC-MS/MS parameters for fusion: peptides were separated over a 120 min gradient, using mobile phase A (0.1% formic acid in water) and mobile phase B (0.1% formic acid in 99% acetonitrile), and eluted at a constant flow rate of 300 nl/min. Acquisition parameters were as follows: data-dependent acquisition with survey scan of 60,000 resolution, AGC target of $4\times10^5$, and maximum injection time (OT) of 100 ms; MS/MS collision induced dissociation in Orbitrap 15,000 resolution, AGC target of $1\times10^5$, and maximum IT of 120 ms; collision energy NCE = 35, isolation window 1.0 m/z. Peak lists were generated in Proteome Discoverer 2.1, and a search was done using Sequest HT (Thermo Scientific) with human Uniprot and fruit bat Uniprot databases. The following search parameters were used: 10 ppm MS; 0.06 Da for MS/MS with the following modifications: oxidation (M) deamidation (N,Q), TMT adduct (N-term, K) carbamidomethyl (C). Peptides detected in WI-38 samples were automatically mapped to their source human protein Uniprot ID. To map the peptides detected in PaLung samples to the corresponding *P. alecto* Uniprot ID, we used a two-step approach. First, we used inParanoid (*O'Brien et al., 2005*) to identify known orthologs between *H. sapiens* and *P. alecto* species. In cases where orthologs were not available, we used blast using the peptide sequence as a query to detect the possible source of *P. alecto* protein. Protein abundances were obtained by summing the abundances of peptides derived from them. We identified differentially expressed proteins by first performing median normalization on all samples from WI-38 and PaLung cells (total six samples), followed by a Student's t-test for all proteins abundances. p-values were corrected using the false discovery rate method of Benjamini-Hochberg.

For the mitochondrial protein-protein interaction network in *Figure 2C*, we first obtained a list of all known mitochondrial proteins from the MitoCarta (*Calvo et al., 2016*) and IMPI databases. These were then input into STRING (*Szklarczyk et al., 2019*) to identify all high-confidence pairwise protein-protein interactions. We used Cytoscape (*Shannon et al., 2003*) to both visualize the network and overlay fold change values of detected proteins onto the network. Clustering of the network and Reactome enrichment of identified subnetworks were performed using the ClusterOne app of Cytoscape. For GSEA, the raw abundances of all samples were used as input and GSEA was run against the same GO BP with pathways gene set list, as with RNAseq data.

## Metabolomics

For the metabolomics comparison, three independent pairs of WI-38 and PaLung cells were cultured using media conditions as described above in 20% O$_2$ and 5% CO$_2$-humidified atmosphere at 37°C before harvesting. WI-38 and PaLung cells were harvested according to the protocol outlined in the

document ACB.1.0.0 provided by Human Metabolome Technologies (HMT Japan). Targeted quantitative analysis was performed by HMT, using capillary electrophoresis mass spectrometry (CE-TOFMS and CE-QqQMS). Absolute abundances (adjusted for cell numbers) were obtained for a total of 116 metabolites (52 and 64 metabolites in the cation and anion modes, respectively). The metabolomics data has been deposited on the Metabolomics Workbench repository (*Sud et al., 2016*).

## Computational flux analysis

### Metabolic network reconstruction

A network model of central metabolism in humans was obtained from Mitocore (*Smith et al., 2017*) (485 reactions, 371 metabolites). This network contains most mitochondrial reactions and pathways, with additional reactions for glycolysis, pentose phosphate pathway, and import and export of amino acids, ions, and other metabolites. The mitocore model also contains a list of genes that catalyze each reaction in a gene-reaction rules table (both mandatory and optional genes). Using this base model, separate context-specific reconstructions for PaLung and WI-38 cell lines were obtained manually using the proteomic and transcriptomic data as follows. The expression levels of all enzymes of the mitocore model were checked in our proteomics and RNAseq datasets, and enzymes were marked as missing if they had <5 counts in two out of three samples (for each PaLung and WI-38) for transcriptomics, and <2000 abundance in two out of three samples for proteomics. Reactions in the mitocore model whose activity depended on the presence of missing genes were iteratively removed from the model while ensuring that the removal of reactions would not result in absence of flux-carrying capacity for ATP production, TCA cycle, or glycolysis. The final reconstructed model for PaLung cells contained 409 reactions and 324 metabolites, and the reconstructed model for WI-38 cells contained 437 reactions and 341 metabolites. We call these PaLung and WI-38 models the 'reconstructed models'.

### Flux sampling

For each species, we first obtained a feasible range of metabolic flux levels for each reaction by performing flux sampling on the 'reconstructed' model for that species. To perform flux sampling, we generated 5000 flux vectors through uniform sampling using the COBRA toolbox v3.0 (*Heirendt et al., 2019*). We call this round of flux sampling the 'control simulation' as there are no constraints imposed on the flux of the reactions for Complex I and oxygen consumption, meaning that these reactions are free to assume any flux values that satisfy the steady-state assumption. Next, we repeated the flux sampling process for a case where the PaLung model was forced to have higher flux through the Complex I reaction than the WI-38 model, and lower flux through the oxygen consumption reaction than the WI-38 model. We call this the 'constrained simulation'. The setting of constraints is explained in the following section.

### Constraints for flux sampling

Constraining reaction fluxes in a metabolic network is usually accomplished by setting a lower bound or an upper bound (or both) on the feasible range of the flux values for individual reactions. When a lower bound is set, flux sampling will only output flux vectors where the flux for that reaction is greater than or equal to the set lower bound. Similarly, setting an upper bound ensures that flux sampling will only output flux vectors where the reaction flux is less than or equal to the set upper bound. To ensure that the PaLung model would have higher Complex I activity and lower mitochondrial oxygen consumption than the WI-38 model, we set flux constraints as follows. We first computed the maximum possible fluxes of the Complex I reaction (Reaction ID: CI_MitoCore) in both PaLung and WI-38 models. We then set the lower bound of the PaLung Complex I reaction flux to a value equal to 70% of its theoretical maximum. Similarly, we set the upper bound of the WI-38 Complex I reaction at a value equal to 30% of its theoretical maximum value. This ensured that the PaLung model would have higher flux through the Complex I reaction, in comparison to the WI-38 model. A similar process of identifying maximum possible fluxes and setting lower and upper bounds was followed for the oxygen consumption reaction (Reaction ID: $O_2$tm) to ensure that the lower bound of the reaction in the WI-38 model was higher than the upper bound in the PaLung model. Flux sampling was then performed for both the constrained PaLung model and the constrained WI-38 model to obtain 5000 flux vectors (as with the control simulation). More details about setting flux sampling constraints and the effects of using different constraint values can be found in Appendix 3.

## MitoSOX assay and antimycin A treatment

PaLung and WI-38 cells were cultured on six-well plates and treated with either DMSO or antimycin A (1 μM) for 16 hr. Antimycin A-treated cells were washed twice with PBS before adding fresh media containing 2.5 μM MitoSOX Red. Cells were stained with MitoSOX Red for 60 min and subsequently collected for flow cytometry. Quantification and analysis of the data were performed by FlowJo software.

## NADPH/NADP+ measurement assay

Intracellular NADPH/NADP+ were measured using the NADP/NADPH Quantification Kit (ab65349, Abcam) according to the manufacturer's instructions. Briefly, $6 \times 10^5$ cells were lysed with 350 μl of extraction buffer. For the reaction, 50 μl of the final sample was used. Signal intensities for NADPH were examined by OD measurements at 450 nm using Infinite M200 plate reader (TECAN).

## GSH/GSSG measurement assay

Intracellular GSH/GSSG was measured using GSH/GSSG-Glo luminescent assay (Promega) according to the manufacturer's instructions. Briefly, $2 \times 10^4$ cells in 96-well plates were lysed in the indicated condition.

## Mitochondrial OCR measurement

OCR was measured using a Seahorse Bioscience XF96 Extracellular Flux Analyzer (Seahorse Bioscience; RRID: SCR_019545). $2 \times 10^4$ cells were plated into Seahorse tissue culture 96-well plates. Cells were cultured in Seahorse assay media containing 10 mM glucose and 2 mM glutamine and incubated in a $CO_2$-free incubator for an hour before measurement. XF Cell Mito Stress Test Kit was used to analyze mitochondrial metabolic parameters by measuring OCR. Oligomycin (1 μM) was injected to determine the oligomycin-independent lack of the OCR. The mitochondrial uncoupler FCCP (1 μM) was injected to determine the maximum respiratory capacity. Rotenone (1 μM) and antimycin A (1 μM) were injected to block Complex I and Complex III of the ETC.

## ECAR measurement

ECAR was measured using a Seahorse Bioscience XF96 Extracellular Flux Analyzer (Seahorse Bioscience). $2 \times 10^4$ cells were plated into Seahorse tissue culture 96-well plates. Cells were cultured in Seahorse assay media containing 2 mM glutamine and incubated in a $CO_2$-free incubator for an hour before measurement. XF Cell Glycolysis Stress Test Kit was used to analyze glycolytic metabolic parameters by measuring ECAR. Glucose (10 mM), oligomycin (1 μM), and 2-deoxy-D-glucose (50 mM) were injected sequentially.

## Acknowledgements

The authors thank Dr. Akshamal M Gamage and Prof. Lena Ho for their insightful commentary on the manuscript. This research is supported by the Singapore Ministry of Education Academic Research Fund Tier 2 grant (MOE2019-T2-1-138) to LTK; Singapore Ministry of Education Academic Research Fund Tier 2 Grant (MOE-T2EP30120-0012) to KI; the Singapore Ministry of Health's National Medical Research Council grant (NMRC/OFIRG/MOH-000639) to KI; and Duke-NUS Signature Research Programme Block Grants to KI and LTK. Any opinions, findings, conclusions, or recommendations expressed in this material are those of the authors and do not reflect the views of the funding agencies.

## Additional information

### Funding

| Funder | Grant reference number | Author |
| --- | --- | --- |
| Ministry of Education - Singapore | MOE2019-T2-1-138 | Lisa Tucker-Kellogg |

| Funder | Grant reference number | Author |
|---|---|---|
| Ministry of Education - Singapore | MOE-T2EP30120-0012 | Koji Itahana |
| National Medical Research Council | NMRC/OFIRG/MOH-000639 | Koji Itahana |
| Duke-NUS Medical School | Block Grants | Koji Itahana Lisa Tucker-Kellogg |

The funders had no role in study design, data collection and interpretation, or the decision to submit the work for publication.

## Author contributions

N Suhas Jagannathan, Conceptualization, Formal analysis, Visualization, Writing – original draft; Javier Yu Peng Koh, Yoko Itahana, Conceptualization, Investigation, Writing – original draft; Younghwan Lee, Investigation, Visualization, Writing – original draft; Radoslaw Mikolaj Sobota, Investigation; Aaron T Irving, Formal analysis, Investigation; Lin-fa Wang, Conceptualization; Koji Itahana, Conceptualization, Writing – original draft; Lisa Tucker-Kellogg, Conceptualization, Formal analysis, Writing – original draft

## Author ORCIDs

N Suhas Jagannathan http://orcid.org/0000-0002-1857-8789
Javier Yu Peng Koh http://orcid.org/0000-0003-4013-9982
Younghwan Lee http://orcid.org/0000-0002-0595-1513
Aaron T Irving http://orcid.org/0000-0002-0196-1570
Lin-fa Wang http://orcid.org/0000-0003-2752-0535
Yoko Itahana http://orcid.org/0000-0003-3560-4560
Koji Itahana http://orcid.org/0000-0002-7241-2894
Lisa Tucker-Kellogg http://orcid.org/0000-0002-1301-7069

## Decision letter and Author response

Decision letter https://doi.org/10.7554/eLife.94007.sa1
Author response https://doi.org/10.7554/eLife.94007.sa2

## Additional files

### Supplementary files

• Supplementary file 1. Genes that pass our differential expression cutoffs (false discovery rate [FDR] < 0.05; |log fold change| > 1) in PaLung vs WI-38 samples from whole-cell transcriptomics data. Differential expression analysis was performed using the DESeq2 pipeline. Fold changes are indicated as PaLung/WI-38.

• Supplementary file 2. Biological pathways upregulated in PaLung cells from gene set enrichment analysis (GSEA) of transcriptomics data. GSEA was performed on the transcriptomics data using (PI) value as a metric. The table below lists the pathways detected as upregulated in PaLung cells (compared to WI-38 cells) and associated enrichment metrics.

• Supplementary file 3. Biological pathways upregulated in WI-38 cells from gene set enrichment analysis (GSEA) of transcriptomics data. GSEA was performed on the transcriptomics data using (PI) value as a metric. The table below lists the pathways detected as upregulated in WI-38 cells (compared to PaLung cells) and associated enrichment metrics.

• Supplementary file 4. Differentially expressed (DE) mitochondrial proteins in PaLung vs WI-38 samples from mitochondrial proteomics data. 405 DE proteins were first identified using a Student's t-test on median-corrected protein abundances from the mitochondrial samples of PaLung and WI-38. Of the 405 DE proteins, 127 were identified to be core mitochondrial proteins (as defined by MitoCarta and IMPI datasets) and are listed in this sheet. Fold changes are indicated as PaLung/WI-38.

• Supplementary file 5. Biological pathways upregulated in PaLung cells from gene set enrichment analysis (GSEA) of proteomics data. GSEA was performed on the proteomics data using protein abundances as input. The table below lists the pathways detected as upregulated in PaLung cells (compared to WI-38 cells) and associated enrichment metrics.

- Supplementary file 6. Biological pathways upregulated in WI-38 cells from gene set enrichment analysis (GSEA) of proteomics data. GSEA was performed on the proteomics data using protein abundances as input. The table below lists the pathways detected as upregulated in WI-38 cells (compared to PaLung cells) and associated enrichment metrics.

- Supplementary file 7. Metabolic model for PaLung cells. A metabolic flux model was constructed for the central carbon metabolism of PaLung cells by overlaying proteomic and transcriptomic information onto the existing mitocore model from literature.

- Supplementary file 8. Metabolic model for WI-38 cells. A metabolic flux model was constructed for the central carbon metabolism of WI-38 cells by overlaying proteomic and transcriptomic information onto the existing mitocore model from literature.

- Supplementary file 9. Flux sampling results comparing flux distributions in the constrained PaLung and WI-38 models. Flux sampling was performed with 5000 flux vectors for the PaLung and WI-38 metabolic models each. The flux histograms for each reaction were compared across the two models and the following statistics were extracted from the histograms.

- Supplementary file 10. Absolute metabolite quantification in PaLung and WI-38 cells. Absolute concentrations of metabolites detected in PaLung and WI-38 cells by Human Metabolome Technologies (HMT).

- MDAR checklist

## Data availability

Transcriptomic data are deposited in the NCBI GEO database (GSE215934). Proteomic data are deposited in the ProteomeXchange database (PXD043121) and in the jPOST repository (JPST001821). Metabolomics data are uploaded to the Metabolomics Workbench database (ST002743). The Matlab script used for flux sampling can be found at here (copy archived at *Jagannathan, 2023*). Other data supporting the findings of this study are available within the article and its supplementary materials.

The following datasets were generated:

| Author(s) | Year | Dataset title | Dataset URL | Database and Identifier |
|---|---|---|---|---|
| Koh J, Irving A, Itahana Y, Lee Y, Itahana K, Wang L, Suhas Jagannathan N, Tucker-Kellogg L | 2024 | RNAseq comparison of lung fibroblasts from Pteropus alecto (PaLung cell line) and *Homo sapiens* (WI-38 cell line) | http://www.ncbi.nlm.nih.gov/geo/query/acc.cgi?acc=GSE215934 | NCBI Gene Expression Omnibus, GSE215934 |
| Itahana K | 2024 | Multi-omic analysis of bat versus human fibroblasts reveals altered central metabolism | https://repository.jpostdb.org/entry/JPST001821 | jPOST repository, JPST001821 |
| Koh J, Jagannathan MS, Sobota RM, Tucker-Kellogg L, Itahana Y, Itahana K | 2024 | Comparison of human and bat metabolism | https://proteomecentral.proteomexchange.org/cgi/GetDataset?ID=PXD043121 | ProteomeXchange, PXD043121 |
| Koh J, Jagannathan NS, Tucker-Kellogg L, Itahana Y, Itahana K | 2024 | Metabolomics comparison of lung fibroblasts from Pteropus alecto and *Homo sapiens* | https://metabolomicsworkbench.org/data/DRCCMetadata.php?Mode=Study&StudyID=ST002743 | Metabolomics Workbench, ST002743 |

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

## Appendix 1

## GSEA of proteomics data from mitochondrial fractions without the outlier P1 and W1 samples

Since the samples P1 and W1 were observed to be outliers from the heatmap in *Figure 2B* of the main text, we repeated the proteomics GSEA, leaving out these two samples and using only P2 and P3 for PaLung and W2 and W3 for WI-38 (*n*=2). However, removing P1 and W1 did not affect our original results and we still observe that ETC and Complex I in particular are upregulated in the PaLung samples.

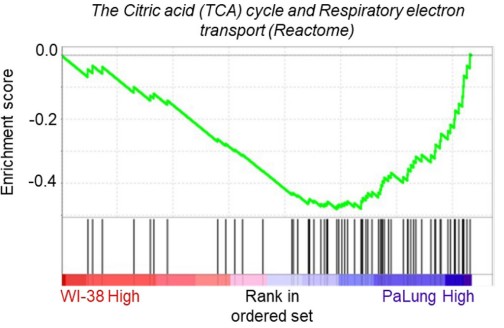
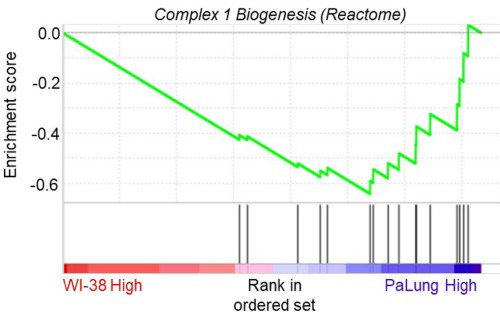

**Appendix 1—figure 1.** Gene set enrichment analysis (GSEA) enrichment plots of the same gene sets as shown in *Figure 2D and E* of the main text, after removing the outlier proteomic samples P1 and W1 from the analysis (*n*=2).

**Appendix 1—table 1.** Table showing the top 35 gene sets enriched in the PaLung mitochondrial proteomics samples, after removing the outlier samples P1 and W1.
Columns indicate the name of the gene set, size (number of genes in gene set), normalized enrichment score, and the false discovery rate (FDR) value.

| NAME | SIZE | NES | FDR q-val |
|---|---|---|---|
| MUSCLE CONTRACTION GOBP GO:0006936 | 35 | –2.301 | 0.002 |
| NICOTINIC ACETYLCHOLINE RECEPTOR SIGNALING PATHWAY PANTHER PATHWAY P00044 | 15 | –2.191 | 0.012 |
| REGULATION OF CELL JUNCTION ASSEMBLY GOBP GO:1901888 | 15 | –2.163 | 0.011 |
| THE CITRIC ACID (TCA) CYCLE AND RESPIRATORY ELECTRON TRANSPORT REACTOME R-HSA-1428517.1 | 64 | –2.154 | 0.011 |
| HALLMARK_OXIDATIVE_PHOSPHORYLATION MSIGDB_C2 HALLMARK_OXIDATIVE_PHOSPHORYLATION | 91 | –2.142 | 0.010 |
| COLLAGEN FORMATION REACTOME DATABASE ID RELEASE 71 1474290 | 31 | –2.096 | 0.015 |
| RESPIRATORY ELECTRON TRANSPORT, ATP SYNTHESIS BY CHEMIOSMOTIC COUPLING, AND HEAT PRODUCTION BY UNCOUPLING PROTEINS. REACTOME R-HSA-163200.1 | 32 | –2.074 | 0.017 |
| RESPIRATORY ELECTRON TRANSPORT REACTOME R-HSA-611105.3 | 32 | –2.059 | 0.019 |
| NADH DEHYDROGENASE COMPLEX ASSEMBLY GOBP GO:0010257 | 16 | –2.059 | 0.017 |
| EPH-EPHRIN SIGNALING REACTOME DATABASE ID RELEASE 71 2682334 | 30 | –2.057 | 0.016 |
| COMPLEX I BIOGENESIS REACTOME R-HSA-6799198.1 | 16 | –2.036 | 0.018 |
| MUSCLE SYSTEM PROCESS GOBP GO:0003012 | 40 | –2.033 | 0.017 |

*Appendix 1—table 1 Continued on next page*

*Appendix 1—table 1 Continued*

| NAME | SIZE | NES | FDR q-val |
|---|---|---|---|
| RHO GTPASES ACTIVATE PKNS REACTOME DATABASE ID RELEASE 71 5625740 | 17 | −2.030 | 0.016 |
| ACTIN FILAMENT-BASED MOVEMENT GOBP GO:0030048 | 17 | −2.022 | 0.017 |
| ACTOMYOSIN STRUCTURE ORGANIZATION GOBP GO:0031032 | 21 | −2.021 | 0.016 |
| MITOCHONDRIAL RESPIRATORY CHAIN COMPLEX I ASSEMBLY GOBP GO:0032981 | 16 | −1.999 | 0.019 |
| INTEGRIN SIGNALLING PATHWAY PANTHER PATHWAY P00034 | 44 | −1.994 | 0.019 |
| KERATINIZATION GOBP GO:0031424 | 15 | −1.976 | 0.021 |
| MITOCHONDRIAL ATP SYNTHESIS COUPLED ELECTRON TRANSPORT GOBP GO:0042775 | 26 | −1.971 | 0.021 |
| MITOCHONDRIAL ELECTRON TRANSPORT, NADH TO UBIQUINONE GOBP GO:0006120 | 15 | −1.968 | 0.021 |
| MITOCHONDRIAL RESPIRATORY CHAIN COMPLEX ASSEMBLY GOBP GO:0033108 | 19 | −1.966 | 0.020 |
| CORNIFICATION GOBP GO:0070268 | 15 | −1.963 | 0.019 |
| SYSTEM PROCESS GOBP GO:0003008 | 77 | −1.953 | 0.021 |
| MIDBRAIN DEVELOPMENT GOBP GO:0030901 | 15 | −1.953 | 0.020 |
| COLLAGEN BIOSYNTHESIS AND MODIFYING ENZYMES REACTOME R-HSA-1650814.3 | 25 | −1.949 | 0.020 |
| KERATINIZATION REACTOME DATABASE ID RELEASE 71 6805567 | 15 | −1.936 | 0.022 |
| ELECTRON TRANSPORT CHAIN (OXPHOS SYSTEM IN MITOCHONDRIA) WIKIPATHWAYS_20191210 WP111 *HOMO SAPIENS* | 28 | −1.933 | 0.022 |
| NABA_CORE_MATRISOME MSIGDB_C2 NABA_CORE_MATRISOME | 34 | −1.921 | 0.025 |
| CELLULAR RESPIRATION GOBP GO:0045333 | 52 | −1.905 | 0.028 |
| HALLMARK_ESTROGEN_RESPONSE_EARLY MSIGDB_C2 HALLMARK_ESTROGEN_RESPONSE_EARLY | 16 | −1.902 | 0.028 |
| EXTRACELLULAR MATRIX ORGANIZATION REACTOME DATABASE ID RELEASE 71 1474244 | 59 | −1.895 | 0.029 |
| RHO GTPASES ACTIVATE PAKS REACTOME R-HSA-5627123.2 | 16 | −1.863 | 0.038 |
| INFLAMMATION MEDIATED BY CHEMOKINE AND CYTOKINE SIGNALING PATHWAY PANTHER PATHWAY P00031 | 30 | −1.848 | 0.043 |
| ELECTRON TRANSPORT CHAIN GOBP GO:0022900 | 31 | −1.842 | 0.044 |
| ATP SYNTHESIS COUPLED ELECTRON TRANSPORT GOBP GO:0042773 | 27 | −1.832 | 0.048 |

## Appendix 2

## GSEA of mitochondrial fractions using mitochondrial gene sets

Our proteomics data was obtained from mitochondrial fraction samples, and the GSEA in *Figure 2* of the main text was originally performed using the entire GO BP gene set list. As an additional measure of redundancy, we repeated our GSEA for the mitochondrial proteomics samples using a gene set list specifically curated for mitochondrial fraction analysis. The gene sets for this analysis were obtained from the MitoCarta 3.0 database (gene set file: MitoPathways3.0.gmx). The results from this analysis also agree with our findings from using the entire GO BP gene set list, that OxPhos and specifically Complex I proteins are upregulated in PaLung mitochondrial samples.

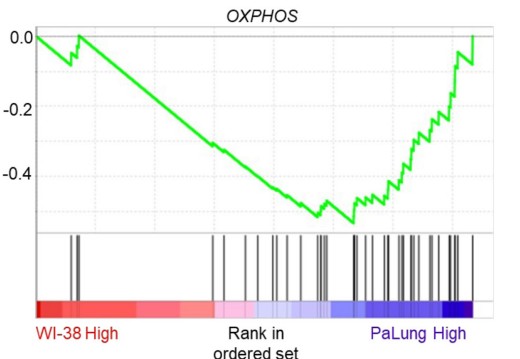 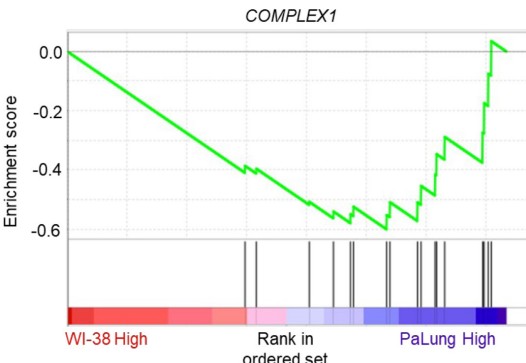

**Appendix 2—figure 1.** Gene set enrichment analysis (GSEA) enrichment plots of the gene sets that are biological equivalents of the gene sets shown in *Figure 2D and E* of the main text. This analysis was performed using the MitoCarta 3.0 gene set list instead of the Gene Ontology Biological Process (GO BP) gene set list used to generate *Figure 2D and E*.

**Appendix 2—table 1.** Table showing the top gene sets enriched in the PaLung mitochondrial proteomics samples, when gene set enrichment analysis (GSEA) was performed using the MitoCarta 3.0 gene set list instead of the Gene Ontology Biological Process (GO BP) gene set.
Columns indicate the name of the gene set, size (number of genes in gene set), normalized enrichment score, and the false discovery rate (FDR) value.

| NAME | SIZE | NES | FDR q-val |
|---|---|---|---|
| OXPHOS_SUBUNITS | 34 | –2.186 | 0.002 |
| OXPHOS | 41 | –2.166 | 0.001 |
| CARBOHYDRATE_METABOLISM | 36 | –2.004 | 0.005 |
| TRANSLATION | 16 | –1.987 | 0.005 |
| COMPLEX_I | 17 | –1.966 | 0.005 |
| CI_SUBUNITS | 15 | –1.892 | 0.007 |
| FATTY_ACID_OXIDATION | 20 | –1.864 | 0.007 |
| METALS_AND_COFACTORS | 30 | –1.794 | 0.011 |
| METABOLISM | 153 | –1.782 | 0.011 |
| AMINO_ACID_METABOLISM | 33 | –1.763 | 0.011 |
| MITOCHONDRIAL_CENTRAL_DOGMA | 24 | –1.609 | 0.030 |
| TCA_CYCLE | 15 | –1.510 | 0.053 |
| LIPID_METABOLISM | 43 | –1.480 | 0.057 |
| PROTEIN_IMPORT_SORTING_AND_HOMEOSTASIS | 24 | –1.200 | 0.223 |
| PROTEIN_HOMEOSTASIS | 18 | –1.053 | 0.377 |

# Appendix 3

## Setting constraints for flux sampling

Our primary goal with the flux simulations is to establish constraints that explore the potential metabolic implications of our key observations from omics and metabolic measurements: bats have upregulated Complex I genes but lower mitochondrial oxygen consumption. Toward this goal, we first set constraints on the Complex I reaction (CI_MitoCore) and the mitochondrial oxygen transport reaction (O$_2$tm) in our metabolic models of PaLung (P model) and WI-38 (W model). In an ideal scenario, these two reactions are completely independent of each other in flux space and setting constraints/thresholds on one reaction does not affect the feasible flux space of the other reaction. However, in our case we observed that constraining one reaction also limits the feasible flux space of the other reaction. Hence, we follow the below protocol in setting our constraints, to ensure that the Complex I flux of bats is higher than humans and vice versa for the O$_2$ flux. We call this the 30-70 protocol as the bounds are set to 30% or 70% of the feasible flux range for each reaction.

i. Compute the minimum and maximum flux possible through CI_MitoCore for the two models. These values are designated as [p_c1_min, p_c1_max] (for PaLung) and [w_c1_min, w_c1_max] (for WI-38).

ii. Constrain the CI reaction in the P model to have a lower bound of p_c1_min + 0.7*(p_c1_max- p_c1_min).

iii. Constrain the CI reaction in the W model to have an upper bound of w_c1_min + 0.3*(w_c1_max- w_c1_min).

iv. Now that the CI reactions have been constrained, compute the new minimum and maximum flux possible through O$_2$ for the two models. These values are designated as [p_o2_min, p_o2_max] (for PaLung) and [w_o2_min, w_o2_max] (for WI-38).

v. Constrain the O$_2$ reaction in the P model to have an upper bound of p_o2_min + 0.3*(p_o2_max- p_o2_min). Designate this upper bound as p_o2_ub.

vi. To avoid overlap of flux ranges and to ensure that the lower bound of the W model O$_2$ reaction is greater than the upper bound of the P model O$_2$ reaction, constrain the O$_2$ reaction in the W model to have a lower bound equal to the higher value between the following two values.
 a. w_o2_min +0.7*(w_o2_max- w_o2_min).
 b. p_o2_ub.

Our methodology may raise the question of what happens if we were to follow the procedure in reverse and first constrain the O$_2$ reaction and then the CI reaction following the same protocol. To answer this question, we performed this simulation as well. As seen in the figure below, this also results in the Complex II reaction having low-to-negative values in the bat model (not seen with the human model).

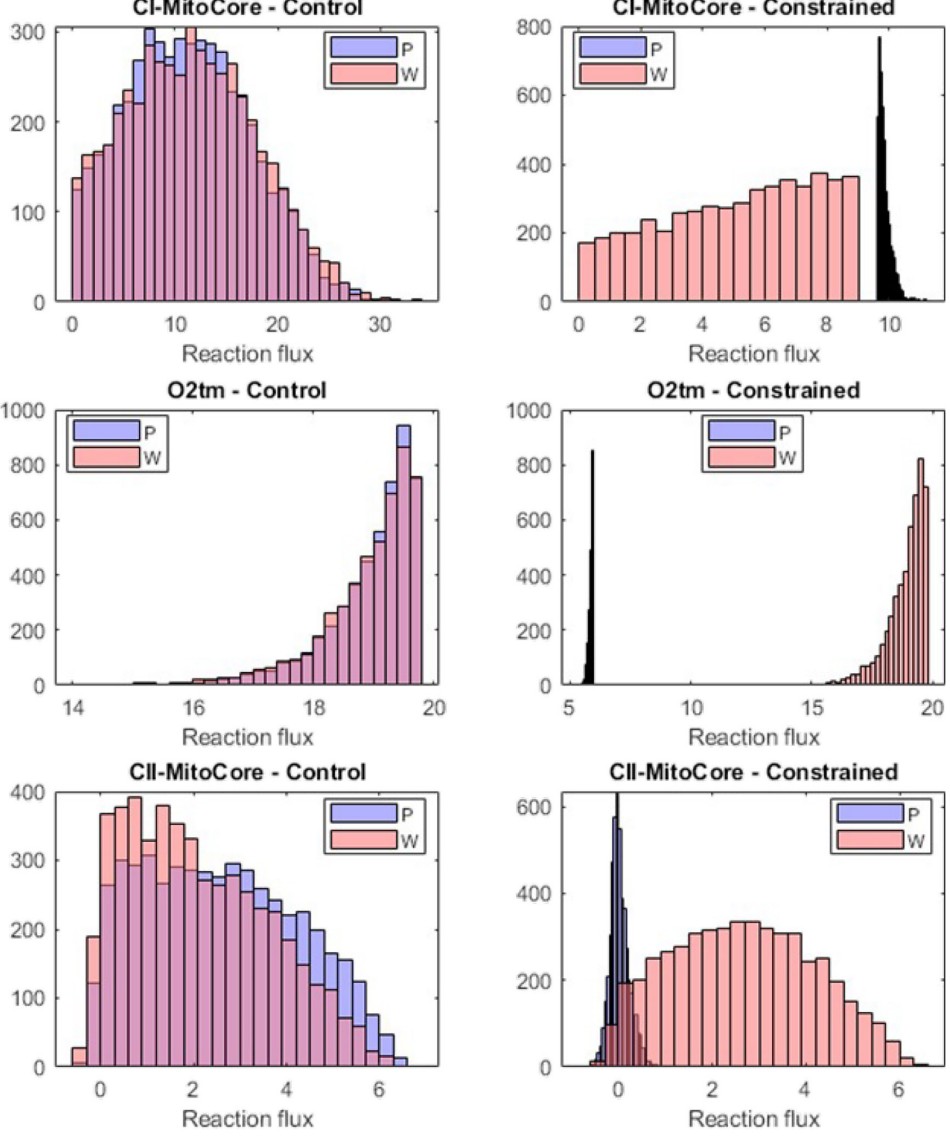

**Appendix 3—figure 1.** Flux sampling histograms of the P and W metabolic models in the unconstrained control (left column) and the constrained (right column) cases. The P fluxes are in blue and the W fluxes in red. The first two rows show the constrained reactions (Complex I and $O_2$), while the third row shows the flux histograms of the Complex II reaction.

In addition, we also provide *Appendix 3—table 1*. *Appendix 3—table 1* shows the minimum and maximum flux possible through the Complex I and $O_2$ reactions when different constraints/combinations of constraints are imposed on the P and W models.

**Appendix 3—table 1.** Minimum and maximum flux values possible for the Complex I and mitochondrial $O_2$ transport reaction when different constraints are applied to the P and W metabolic models.

| | Complex 1 | | | | Oxygen transport | | | |
| | Bat (P model) | | Human (W model) | | Bat (P model) | | Human (W model) | |
| Constraint description (all follow the 30-70 protocols) | Min | Max | Min | Max | Min | Max | Min | Max |
| --- | --- | --- | --- | --- | --- | --- | --- | --- |
| Control simulation – no constraints | 0 | 41.43 | 0 | 41.44 | 0 | 19.8 | 0 | 19.8 |

*Appendix 3—table 1 Continued on next page*

*Appendix 3—table 1 Continued*

| Constraint description (all follow the 30-70 protocols) | Complex 1 | | | | Oxygen transport | | | |
| --- | --- | --- | --- | --- | --- | --- | --- | --- |
| | Bat (P model) | | Human (W model) | | Bat (P model) | | Human (W model) | |
| | Min | Max | Min | Max | Min | Max | Min | Max |
| Ideal target flux range expected with the 30-70 protocol, when CI and $O_2$ are independent of each other | 29 | 41.43 | 0 | 12.43 | 0 | 5.94 | 13.86 | 19.8 |
| Constraining only the Complex I reaction | 29 | 41.43 | 0 | 12.43 | 13.58 | 19.8 | 0 | 19.8 |
| Constraining only the $O_2$ reaction | 0 | 13.74 | 0 | 41.43 | 0 | 5.94 | 13.86 | 19.8 |
| Constraining CI first, then constraining $O_2$, without avoiding flux range overlap | 29 | 32.72 | 0 | 12.43 | 13.58 | 15.45 | 13.86 | 19.8 |
| Constraining CI first, then constraining $O_2$, avoiding flux range overlap | 29 | 32.72 | 0 | 12.43 | 13.58 | 15.45 | 15.45 | 19.8 |
| Constraining $O_2$ first, then constraining CI, without avoiding flux range overlap | 9.61 | 13.73 | 0 | 12.43 | 3.88 | 5.94 | 13.86 | 19.8 |
| Constraining $O_2$ first, then constraining CI, avoiding flux range overlap | 9.61 | 13.73 | 0 | 9.61 | 3.88 | 5.94 | 13.86 | 19.8 |

We also performed further flux sampling simulations to explore the effect of using the 30-70 protocol for threshold percentages, and whether the results would be robust to other threshold percentages. We repeated our original simulations following our original protocol (constrain C1 first and then $O_2$), with thresholds of 20-80, 40-60, and 50-50 *Appendix 3—figure 2* . In all cases, we observed that the Complex II reaction has low-to-negative values in the bat model compared to the human model to different degrees. We thus conclude that this observation is robust to the choice of constraint thresholds.

Our flux sampling scripts can be found at https://github.com/narendrasuhas/PalungWI38FluxSim, (copy archived at *Narendrasuhas, 2024*). The script contains a variable called minFrac, which can be set to values between 0 and 1 for setting different thresholds. For example, a minFrac value of 0.3 ensures a 30-70 protocol, while a minFrac value of 0.2 ensures a 20-80 protocol.

## Unconstrained

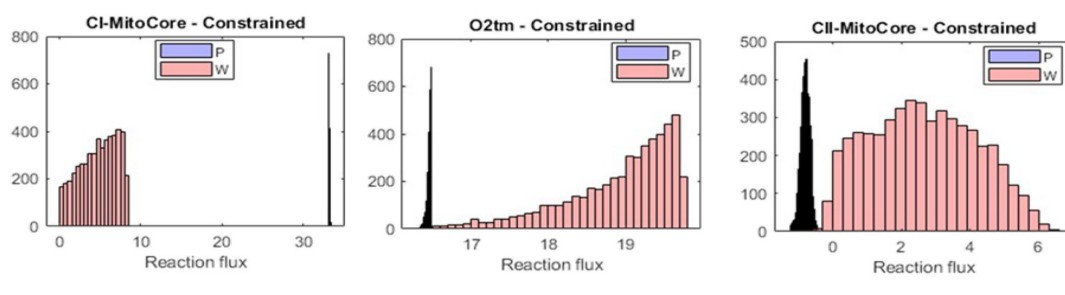

## Constraints: 20-80 protocol

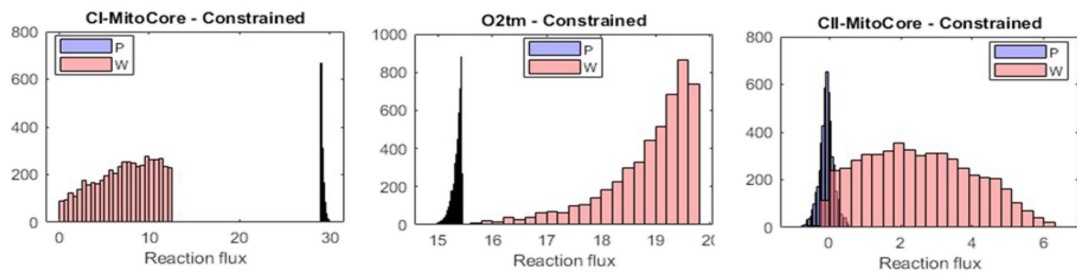

## Constraints: 30-70 protocol (Used in main text)

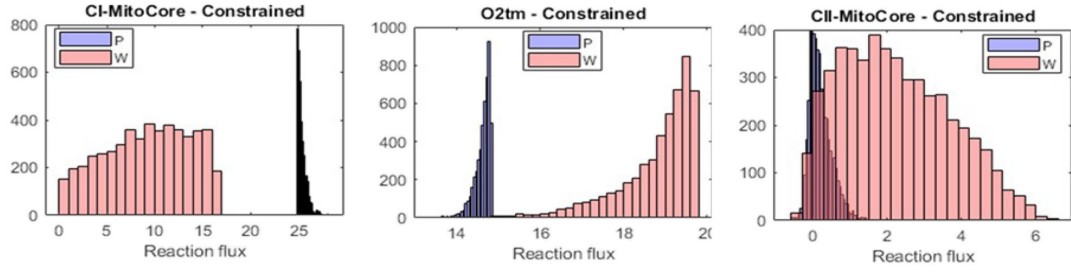

## Constraints: 40-60 protocol

## Constraints: 50-50 protocol

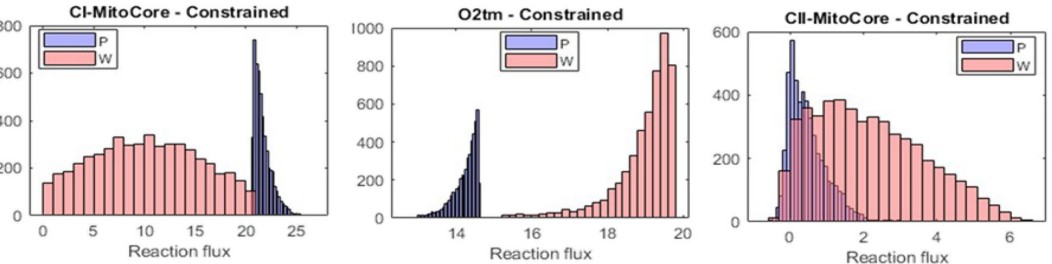

**Appendix 3—figure 2.** Flux sampling histograms of the P and W metabolic models under different threshold values for constraints. Each row corresponds to a constraint scheme (unconstrained, 20-80, 30-70, 40-60, and 50-50), and was obtained by running our script with different minFrac values. Within each panel, the P model
*Appendix 3—figure 2 continued on next page*

*Appendix 3—figure 2 continued*

fluxes are in blue and the W model fluxes in red. The leftmost column shows the flux histograms of the Complex I reaction, the middle column shows flux histograms of the mitochondrial $O_2$ transport reaction, and the rightmost column shows the flux histograms of the Complex II reaction.

