## [Editor Report]

This study analyzed the metabolism of bat cells versus human cells through a comprehensive multi-omics approach, focusing on the black flying fox fruit bat. Findings revealed that bat cells have higher expression levels of Complex I in the electron transport chain but a lower oxygen consumption rate, suggesting a unique metabolic state similar to ischemia. Despite higher levels of mitochondrial reactive oxygen species, bat cells displayed greater antioxidant reserves and resilience to metabolic stress, including glucose deprivation and ferroptosis, highlighting fundamental metabolic differences supporting bats' increased longevity and disease resistance. The study is compelling and provides solid evidence to back the hypothesis.

---

## [Decision Letter]

[Editors' note: this paper was reviewed by Review Commons.]

---

## [Author Response]

1. General Statements [optional]

We thank all three reviewers for their valuable comments and sincerely appreciate the diligence and their attention to detail in reading our manuscript “Multi-omic analysis of bat versus human fibroblasts reveals altered central metabolism.” All three reviewers agreed that our work is significant, and reviewers described the strengths of our work as follows: novel insight into bat metabolism, providing a first-of-its-kind comprehensive study using multi-omics approaches and modelling for inter-species comparison, and providing a valuable stepping stone for further research on bat metabolism.

The reviewers also raised individual questions about specific points in our work and made suggestions for improving the strength of our claims. We agree with the reviewers' assessments in most cases and will incorporate their suggestions into the text, analyses, and interpretation of our work. Our point-by-point responses explain how we will address each of the reviewers’ concerns, on both the wet bench and computational aspects of our work. These include redoing the following: glucose deprivation cell death assay, transcriptomics analysis using current bioinformatics pipelines, GSEA analysis for both transcriptomics and proteomics data, and flux sampling using modified constraints. Several requests for additional information have been addressed by providing the relevant rationale/justification in our responses. Reviewer 2 suggested biochemical validation of the upregulation of Complex I in bat cells. However, it remains challenging to perform such validation using qPCR or Western blotting in the bat species for reasons that are detailed in our responses. Therefore, following Reviewer 1's suggestion, we have decided to moderate our claims in this manuscript.

2. Description of the planned revisionsReviewer 1 Major comments1. Regarding Figure 1A, the authors mention 'n = 3' for a single cell line. Does this refer to three different passages or three independent experiments? Please provide a more detailed description to clarify.

We would like to thank the reviewer for the comment. We apologize for not providing sufficient descriptions of the cells and experiments. Our datasets in Figure 1 were generated from three independent experiments, comparing the cell lines WI-38 (W) and PaLung (P) respectively: W1 vs P1, W2 vs P2, and W3 vs P3. These sets of cells were harvested at different passages in each experiment. Cells were generally subcultured at a 1:2 ratio and they were collected at passage numbers 26 (W1), 30 (W2), and 35 (W3) or 5 (P1), 6 (P2) and 7 (P3). We will provide these details in the Material and Methods section.

2. In relation to Figures 1C and 1D, the authors state in the figure legend that the 'GSEA analysis identifies Respiratory electron transport and Cellular response to hypoxia as the top metabolic pathways that are differentially regulated between PaLung and WI-38 cells.' (Lines 140-144). However, the criteria for selecting these terms as the top metabolic pathways is not clear. In the lists in Supplementary Tables 2 and 3, the authors' proposed term, 'Respiratory electron transport,' is ranked 126th, and 'Cellular response to hypoxia' is ranked 79th. Conversely, terms related to the TCA cycle are ranked 66th and 82nd, and another term that seems to be related to hypoxia, 'OXYGEN-DEPENDENT PROLINE HYDROXYLATION OF HYPOXIA-INDUCIBLE FACTOR Α,' is ranked 62nd. Could the authors please provide a clarification for their choice of 'Respiratory electron transport' and 'Cellular response to hypoxia' as the top metabolic pathways?

We thank the reviewer for raising this question. When presenting results in our original submission, we chose to display gene sets that satisfied the following requirements –

1) The gene set pertains to central metabolism (defined as being a sub-gene set of the superfamily Cellular Metabolic Process GO:0044237)

2) The gene set satisfies the FDR < 0.25 threshold commonly used for GSEA analysis (as opposed to p < 0.05 used in conventional statistics). Justification for why a higher FDR threshold is used for GSEA can be found on the GSEA FAQ website.

(https://software.broadinstitute.org/cancer/software/gsea/wiki/index.php/FAQ)

3) Has a nominal enrichment score with absolute value >= 1

4) When multiple similar gene sets (by biological pathway) satisfy the above constraints, we pick the largest gene set (by gene count) to get a more high-level comparison of metabolism

It was from these criteria that we originally displayed the respiratory electron transport gene set (over the similar TCA cycle gene sets, although the TCA gene sets were ranked higher), and the cellular response to hypoxia gene set. In response to this reviewer’s later comments about the bioinformatics pipeline used (minor comment 1), we propose to redo our RNAseq analysis using updated methods such as STAR. We will also redo our GSEA using the newly updated results from the improved pipeline and will follow the same steps listed above to identify the core metabolic processes that are altered in PaLung cells. When describing these new results in the manuscript text, we will point out explicitly that the GSEA pathways chosen for display/discussion are not the absolute top pathways shown by GSEA, but among the top few metabolic gene sets and were selected by this protocol for display. In addition, we will also add supplemental material showing ES plots similar to the ones in Figure 1 for other similar gene sets that may be ranked higher in the GSEA ranking than the ones displayed in Figure 1.

3. In the Materials and methods section (lines 419-421), the authors mention, 'GSEA was run against the complete Gene ontology biological process (GO BP) gene set list (containing 18356 gene sets).' However, they narrow down the gene dataset for analysis (lines 136-138, 'we filtered our gene dataset to contain only genes listed under the Gene ontology category Cellular Metabolic Process (GO ID:0044237), resulting in a truncated list of 4794 genes.'). I'm concerned that this selective approach might introduce bias into the resultant pathways. Is this selective approach commonly employed in this type of analysis? And isn't there a need for adjustments to avoid potential bias?

We thank the reviewer for identifying this error on our side. It is true that in our initial analysis, we ‘filtered our gene dataset to contain only genes listed under the Gene ontology category Cellular Metabolic Process (GO ID:0044237), resulting in a truncated list of 4794 genes.’ Hence this line was included in the original version of the manuscript. However, owing to the same concerns raised by this reviewer, we later decided against this approach and ran the GSEA analysis using ALL genes detected in our RNAseq experiments (without filtering for metabolic genes). Figures 1C-D, and Supplementary Tables 2-3 were all generated from this GSEA analysis with the complete gene set list. However, it was an oversight on our part to not amend the manuscript text in lines 136-138. We propose to remove the corresponding lines in the revised manuscript.

4. The authors noted that the number of differentially expressed genes (DEGs) is quite high (6,247 out of 14,986) as per lines 134-135, stating that "The number of differentially expressed genes (6,247) was extremely high, suggesting that multiple pathways are differentially regulated between the two species." However, this large number of DEGs could indicate either an improper correction procedure or a need for a more stringent threshold. The authors should address this issue to avoid potential misinterpretation of the results.

We agree with this suggestion. The words “differentially expressed” are problematic in an inter-organism comparison. In our original submission, we never used the 6247 “differential expressed” genes as a pre-filter for downstream analysis, and our GSEA analyses used ALL genes common to both organisms. For the revised manuscript, we will refrain from using the term differentially expressed in the transcriptomic context. As suggested by reviewer 3, we will simply refer to them as genes that pass our cutoff threshold. In addition, we will also make our thresholds more stringent (*p* < 0.01 and |log fold change| > 1.5).

In terms of the correction procedure, in addition to the typical normalization performed, we also do correct for differences in transcript/gene length in different organisms as part of our analysis, and we believe this correction should suffice for inter-species comparison of RNAseq experiments.

5. In Figure 2B, the samples labeled as W1 and P1 appear to be outliers. This raises questions about the integrity of the sampling or analysis process. Please describe about this.

We would like to thank the reviewer for the comment and concern. Indeed, proteomics was done in pairs of PaLung and WI38 samples on different days, (P1, W1), (P2, W2), (P3, W3). It is possible that there might have been batch effects in the first pair that caused them to be outliers. Therefore, in addition to the main figure (n = 3), we will redo the proteomics analysis, including GSEA, after discarding the pair (P1, W1) and include this additional analysis in the supplementary figures.

6. Regarding the GSEA analysis of Figure 2, they are using the full set of GSEA. However, this reviewer is wondering if this is appropriate when analyzing mitochondrial fractions, as I believe using the entire GSEA set could introduce a bias. Is this a common approach? Shouldn't the authors be focusing on mitochondrial-related sets within the GSEA, and then determining the upregulated and downregulated pathways from there?

In response to this reviewer’s question, we propose to redo the GSEA analysis of the proteomics data using gene sets that are more specific for mitochondria. An example can be found in the Mitocarta database (https://www.broadinstitute.org/mitocarta/mitocarta30-inventory-mammalian-mitochondrial-proteins-and-pathways), which contains a gene set file (MitoPathways3.0.gmx). This gene set file is tailored for GSEA and contains 1136 mitochondrial genes sorted into 149 mitochondrial pathways/gene sets.

7. The authors describe in lines 195-197, "GSEA-flagged upregulation in OxPhos was driven mostly by the upregulation of Complex I subunits, for both the proteomic and transcriptomic data (Figure 2G, Supplementary Figure S1D)." However, within this analysis, the number of genes composing each subgroup of the mitochondrial Complexes are 44 for Complex I, 4 for Complex II, 10 for Complex III, and 19 for Complex IV (https://www.genenames.org/data/genegroup/#!/group/639). The authors mention that the genes of Complex I were dominant in the ETC, but, might this just be reflecting the original difference in the number of genes? As this reviewer believes this could have a significant impact on the authors' current claims, this reviewer suggest the authors to carefully reconsider this point, comparing the actual results with the proportion expected from the difference in gene numbers. (Even in Figure S1D, it appears to correlate with the number of genes: C1 39.3%, C3 10.7%, C4 10.7%, C2 3.5%)

Combined response for points 7 and 8 below.

8. As pointed out in Major Point 7, if the authors' claim of enrichment in Complex I is indeed due to the large number of genes included in the Complex I subgroup (https://www.genenames.org/data/genegroup/#!/group/639), can the assumption of High Complex I flux truly be considered valid? In that case, this constraints model would become inappropriate, and the validity of the inferred low or reverse activity of Complex II would be diminished. Therefore, a careful re-examination is desirable.

We interpret points 7 and 8 as having distinct but related concerns. (1) Is the enrichment of ETC just an artifact of Complex I having many subunits? (2) Is the ETC gene set enriched because of upregulation Complex I, or because of multiple mitochondrial complexes and it only appears to be dominated by the Complex I significance due to Complex I having many subunits? (3) Is the enrichment of Complex I just an artifact of Complex I having many subunits? Responses to these concerns are below.

In the first concern, the reviewers ask if it’s true that the proportion of observed Complex I genes (out of all possible Complex I genes) is larger than the proportion of observed Complex II, Complex III or Complex IV genes (compared to all possible Complex II, Complex III, and Complex IV genes), in the final GSEA core enrichment set. The reviewer suggested we compare the actual results versus the proportion expected from the difference in gene numbers. Therefore, we performed the following statistical analysis:

The null hypothesis is that the number of Complex I-IV genes observed in the core enrichment set match the frequencies of genes expected from the number of subunits in each complex. For transcriptomics, a chi-square test of goodness-of-fit for the observed counts of ETC genes in the core enrichment set vs expected counts based on gene proportions (Complex I: 44; Complex II: 4; Complex III: 10; Complex IV: 19) fails to reject the null hypothesis (that the observed counts are drawn from the expected frequencies) with a *p*-value of 0.6419 (chi-square statistic = 1.6779; dof = 3). The failure to reject the null hypothesis means that the actual results are similar to the expected frequencies by proportion.

This is no contradiction with our statements, because we do not claim that Complex I is preferentially enriched in PaLung in comparison to other ETC complexes. In fact, we do not make any claims or assertions about the other complexes of the ETC such as Complex II, Complex III and Complex IV from a transcriptomic or proteomic perspective.

To address the second concern, we will rephrase our text slightly, because we don’t mean to make a claim that Complex I is the only reason why the ETC gene set is enriched. Our claim is only that due to the absolute number of genes in Complex I (which is proportional to expected frequencies), significant enrichment of Complex I alone may be sufficient for GSEA to call the entire ETC gene set upregulated, without requiring a strong signal-to-noise ratio of enrichment due to the other complexes. We will modify the text in our manuscript to clarify this issue.

For the third concern, our claim of upregulated Complex I is substantiated by the fact that in proteomics, gene sets specific to Complex I are upregulated in PaLung. This observation is independent of other ETC Complex genes.

In summary, our claims are the following,

It is true that Complex I-specific gene sets are upregulated in PaLung cells. This underlies our assumption of higher flux through Complex I for flux modelling.It is possible that these upregulated Complex I genes may be causing the broader ETC gene set to be flagged as upregulated.It is not possible to infer from this analysis alone whether Complex II, Complex III and Complex IV are also up- or down-regulated, because individual gene sets for these complex activity/assembly were not flagged as up- or down-regulated in our analysis (unlike with Complex I).

9. (option, takes about 1-2 months). This reviewer believes that the authors' most important claim, concerning the high activity of Complex I and the low activity of Complex II, lacks strong evidence as no biochemical data of the activities of each mitochondrial complex are presented to substantiate this. Unless additional biochemical experimental data is provided, the assertions should be toned down. While the abstract mentions "complex II activity may be low or reversed," it is stated with certainty in line 108 of the introduction, "associated with the low or reverse activity of Complex II." Based on the present data, this reviewer believes that the claim remains speculative. Therefore, I suggest moderating the overall argument or adding the biochemical data. While the results from metabolomics are supportive, they do not serve as direct evidence.

We agree, and we thank the reviewer for noticing that certain locations have a higher strength of assertion. We will change the flagged line (Line 108) and moderate our assertion about possible low-to-reverse activity of Complex II and avoid over-interpreting the data.

10. Regarding Figure 5, the title of the figure states "lower antioxidant response", but it doesn't seem that the data in the figure actually shows a lower antioxidant response.

We thank the reviewer for noticing this discrepancy. We propose to modify the title of Figure 5 to “ROS and antioxidant system measurements in PaLung and WI-38 cells”.

11. In lines 109-110 of the Introduction, the authors state, "we confirmed our prediction of ischemic-like basal metabolism in PaLung cells by characterizing the response of bat cells to cellular stresses such as oxidative stress, nutrient deprivation, and a type of cell death related to ischemia, viz. ferroptosis." However, can the assertion that the cells are in an ischemic-like state be confirmed simply because they are resistant to several types of cellular stress?

We will moderate our language to clarify that resistance to nutrient deprivation and cellular stress are consistent with an ischemic-like state, but do not confirm an ischemic-like state. This change applies to both the results & Discussion sections.

Reviewer 1 Minor comments1. The authors mention the use of cufflinks/Tophat for mapping/quantification. However, support for these software programs has ended and the creators of these programs themselves recommend using the successor programs. I recommend re-analysis using a more current pipeline (such as HISAT2/StringTie, STAR/RSEM, etc.). Furthermore, the transcriptomics section of the methods should also include the program used for cleaning and trimming.

We thank the reviewer for bringing our attention to this issue. For our revision plan, we propose to redo the transcriptomic analysis with the newer pipelines suggested by the reviewer and update our RNAseq results in the manuscript text, figures, and supplementary material as required.

2. As for the Oxygen Consumption Rate (OCR) data presented in Figure 2F, it makes sense that it's low at the basal level. However, it's perplexing that it is also low even under uncoupled conditions, especially considering the high energy demand associated with flight in this species. Could the authors provide their interpretation on this apparent contradiction?

We would like to thank the reviewer for the constructive and insightful comments. We also did not expect that bat cells would have a lower capacity than WI-38 cells under uncoupled conditions (which indicate maximal respiratory capacity). Given that flight is an energetically demanding activity (Maina JN 2000), bat cells are expected to have a high oxygen consumption capacity. Future research can test one possible interpretation, which is that this phenomenon might be cell-type dependent. For example, bat muscle cells, which possess numerous mitochondria and are the primary drivers of flight, might have a higher maximal respiratory capacity than human muscle cells. In contrast, bat fibroblasts, which might play a less important role in flight, might not require such a high capacity. Additionally, *P. alecto* can hibernate. To endure the challenging conditions of hibernation, bat fibroblasts may have adapted to reduce their oxygen intake and limit energy use. Because further analysis of different cell types will be essential to pursue these issues, we will incorporate them into the Discussion section. We hope this will be useful for future research to better understand the cell biology of bats.

Reference

Maina JN. What it takes to fly: the structural and functional respiratory refinements in birds and bats. J Exp Biol. 2000, 203:3045–3064, doi: 10.1242/jeb.203.20.3045

3. In line 156, the authors mention that 'Profiling detected a total of 1,469 proteins.' Please provide more details in the explanation. Specifically, does this total of 1,469 proteins represent a combined count from both humans and bats, or is this the number of proteins for which orthologs could be identified in both species, just like the authors did with the transcript results.

We will clarify that “The list of 1,469 proteins is composed of all proteins whose peptides were detected with high confidence in both species. There were no peptides detected in our experiment that were exclusively detected in high confidence in only one organism”. Further explanation in the methods section will indicate that orthologs for all 1469 proteins were identified using a combination of inParanoid and BLAST.

4. In Supplementary Table 4, only 127 mitochondrial proteins are listed out of the 405 proteins mentioned in "Of these 405 proteins, we identified 127 to be core mitochondrial proteins (lines 161-163)". As there is no explanation for this within Supplementary Table 4, it would be better to include one.

Thank you for this comment. In our revised manuscript, we shall update this table to contain data about all 405 mitochondrial proteins and not only the 127 proteins that exceeded our differential expression cutoffs.

5. In line 472, the phrase "GO BB gene set list" is used. Could this potentially be a typographical error, and should it instead be "GO BP gene set list"?

This was a typo. It should be the GO BP gene set list and will be corrected in the revised manuscript.

6. In the volcano plot of Figure S3B, it appears that the side with lower P/W values generally corresponds with lower p-values. I wonder if there might have been any oversight or mistake in the data analysis process that could explain this observation?

We thank the author for catching this puzzling asymmetry. It is true that in the volcano plot in S3B, many proteins that are high in WI-38 and low in bats (with a low P/W ratio) happen to fail statistical significance. To delve deeper into this issue, we extracted four subsets of the total set of 1469 proteins as follows based on their p-value (p) and log fold change (LFC).

Subset 1 (323 proteins): Low log(p) and low LFC (p > 0.05 and |LFC| < -1). This is the set of proteins that are highly enriched in the WI-38 sample but do not pass statistical significance.Subset 2 (118 proteins): High log(p) and low LFC (p <= 0.05 and |LFC| < -1). This is the set of proteins that are highly enriched in the WI-38 sample and pass statistical significance.Subset 3 (143 proteins): Low log(p) and high LFC (p > 0.05 and |LFC| > 1). This is the set of proteins that are highly enriched in the PaLung sample and do not pass statistical significance.Subset 4 (289 proteins): High log(p) and high LFC (p <= 0.05 and |LFC| > 1). This is the set of proteins that are highly enriched in the PaLung sample and pass statistical significance.

For both PaLung and WI-38 samples and for each subset, we looked at the histogram of coefficient of variation (CV; standard deviation/mean) for the absolute abundance of each protein (See Author response image 1). Compared to the other histograms, we found that the CV histogram for the WI-38 samples in subset 1 had a more uniform distribution and included much higher values. Delving deeper into WI-38 subset 1, we identified 180 proteins that had a CV value greater than 0.7, suggesting that these proteins showed either high standard deviations or low mean abundance. Plotting the mean log(abundance) for the WI-38 samples across all four subsets, we found that subset 1 did not have lower abundances than the other subsets (Author response image 2). The abundance distribution for the 180 high-CV proteins also matched the abundance distribution for the rest of subset 1. All of this suggests that the observed asymmetry in the volcano plot is caused by a high standard deviation in 180 proteins in the WI-38 sample, which could be a batch and species-specific effect of metabolic regulation.

**Author response image 1. sa2fig1:** Histograms of coefficient-of-variation (CV) for the four protein subsets (defined by p-value and log fold change). Each row corresponds to a different subset indicated by the title above. Within each row, the blue histogram corresponds to the CV from the 3 PaLung samples, and the red histogram corresponds to the CV from the 3 WI-38 samples. The one major discrepancy is that the histogram for the WI-38 samples in subset 1 (Row one right panel), contains much higher values than the other 7 histograms.

**Author response image 2. sa2fig2:** Histograms of log(abundance) for the four protein subsets (defined by p-value and log fold change) for WI-38 samples. Each panel corresponds to a different subset indicated by the title above.

7. In lines 249-252, it is stated, "The low or negative flux values for Complex II in our PaLung simulations indicate that the electrons obtained from Complex I may accumulate at Complex II or potentially even get consumed by Complex II operating in reverse (bypassing the rest of the ETC) in PaLung cells." However, isn't the basic process of electron transfer done through Complex I-III-IV, independent of Complex II?

We thank the reviewer for raising this point. It is true that in conventional ETC, both Complex I and Complex II operate in parallel and feed electrons to Complex III and downstream. However, in cases of dysregulated ETC observed with the reverse activity of Complex II, it is possible that the electrons generated by Complex I do not proceed to Complex III and could be utilized by Complex II for its reverse activity (Bisbach et al., Cell reports 2020; Chouchani et al., Nature 2014). This phenomenon has been pointed out by earlier studies, two examples of which are shown below. In place of our earlier sentence, we will amend our manuscript text to say the following-

“During conventional ETC, both Complex I and Complex II operate in parallel and produce electrons that are shuttled downstream t to Complex III, Complex IV, and ATP synthase. However, prior work has documented an alternative in which the electrons obtained from Complex I can be consumed by Complex II operating in reverse, rather than traversing the rest of the ETC. This alternative was utilized by the computational models that showed low or reverse activity of Complex II.

8. Regarding Figure 4F, the authors state, 'PaLung cells displayed higher viability than WI-38 cells after glucose deprivation (Figure 4F).' However, in addition to the cell images, it would be beneficial to perform experimental quantification of cell death to provide more rigorous data. Additionally, the cells appear to be over-confluent, which might influence the results. Also, scale bars should be included in all photos, including Figure 6.

We are grateful for the valuable suggestions and feedback from the reviewer. For Figure 4F, we will repeat the experiments to quantify cell death at the same cell density as in Figure 6A and C. We will avoid the over-confluent conditions and include the scale bars in all the photos.

9. Regarding Figure 5B, it is stated that 'the expression levels of differentially expressed antioxidant genes' are shown, but it includes those that are not significant. It would be helpful if the authors could clarify how this gene set was selected.

Thank you for catching this omission in our explanation. The genes included in Figure 5B were only those that showed statistically significant differences. Asterisks are missing for three of the proteins. Since we will redo the transcriptomic analysis, the revised figure will have minor differences in the counts and TPM, and we will also correct the annotations in the updated figure. To address possible concerns about the set of antioxidant genes, we will plot all genes in the HALLMARK_REACTIVE_OXYGEN_SPECIES_PATHWAY set that are statistically significant in expression difference between PaLung and WI-38 cells.

10. Regarding Figure 6C, the values for total glutathione seem to significantly differ from those in Figure 5C. An explanation for this discrepancy would be appreciated to ensure the consistency and reliability of the data.

We thank the reviewers for the careful and critical reading of our manuscript. We apologize for our careless mistake. We noticed that the values in Figure 5C were normalized by protein amount, but Figure 6F was not. We will correct the labelling of the y-axis and replace Figure 6F with a graph showing the total glutathione normalized by protein amount (nmol/mg protein). After normalization, the values in Figure 6F are similar to those in Figure 5C.

Reviewer 2 Major comments1. The authors compared a fibroblast cell line derived from adult bats with a human embryonic cell line. Please discuss whether mitochondrial metabolism in embryonic cells might be different and how it could have affected the obtained results. Please describe in more detail how the cells were established, what population doubling they were used at (both bat and human cells). Were the cells cultured in atmospheric oxygen or low-oxygen conditions. The exposure of cells to atmospheric oxygen might affect the many mitochondrial parameters measured in this study and could influence the main finding about ischemic-like state. Additionally, please mention in the limitations of the study that only biological n=1 was compared (since cells only from 1 individual per species was used in experimental groups), despite n=3 technical replicates.

We would like to thank the reviewer for the help in improving the quality of our work. We will make the necessary revisions to ensure that our Materials and methods section is clear and complete, and that we address the limitations of our study in the Discussion section.

Regarding the comparison of fibroblast cell lines from adult bats and human embryo stage, we acknowledge the potential differences in mitochondrial metabolism between these cell types. In the revised manuscript, we will discuss the differences in more detail, using the existing literature on mitochondrial metabolism.

We will describe the methods used to establish and propagate these cell lines, including the population doubling at which they were utilized in our experiments. We cultured them in a normoxic condition (not in a hypoxic condition). We will mention in the Discussion section that the exposure of cells to different oxygen levels could affect the metabolic phenotypes.

We appreciate the reviewer's comments that our study used cells from only one individual of each species, which is a major limitation. We will explicitly mention this in the Discussion section to ensure that readers are aware of the limitations of the sample size.

2. Reference genomes for bats are not as well annotated as for human. Downregulation of a pathway may result from some genes being excluded from the analysis because of poor annotation of the P. Alecto genome compared to human. The authors state: "Genes with counts per million (CPM) < 1 in more than 3 out of 6 samples were discarded from downstream analysis". So, if the gene was not annotated, was it assigned a zero value and discarded? Was it discarded if it was zero in one species (e.g. bat) or set to 0? If such genes were excluded, while in reality not being mapped, they could have skewed the pathway analysis.

The reviewer raises an important point here. It is true that reference genomes for less studied organisms like bats may not be as well annotated as other organisms. Indeed, the seeming down-regulation of a pathway in bats might be because the constituent genes were not annotated as well, not because they weren’t expressed as highly. For exactly this reason, we analysed pathways that were upregulated in bats, and made no conclusion about pathways downregulated in bats. We will make this more clear in our text.

For the question of discards, the reviewer’s interpretation is correct. Our transcriptomic comparison was performed only using genes/transcripts common to both organisms. If a gene was not detected in the bat experiment, it was removed from the comparison. We will clarify in the discussion that this discard would exclude genes that are unmapped in bat, and the pathway analysis might under-estimate bat-specific biology.

4. The major findings of this paper were based on the omic data, followed by some experimental validations. However, the quality of these omic data or the results are not solid enough to motivate the authors to validate these findings. For example, both of the GO terms enriched by the DEGs in Figure 1 are not the top terms as claimed by the authors (not even significant after multiple test correction). Also, even though the 2 GO terms in Figure 2 are quite significant, the expression pattern seems not very consistent among the replicates, which make the enrichments not so solid. This highlights an inconsistency among different omic datasets, which may generate some conflicting results. For example, the low level of metabolites from TCA cycle (Figure 4c) seems not consistent with the high level of TCA-related protein, as described in Figure 2c & d. For the purpose of improving the manuscript quality, the authors may have to evaluate the consistency among the multiple omic datasets or to optimize their bioinformatic pipeline to enhance the results.

Regarding the choice of GO terms, we agree this should be cleaner. When we redo our transcriptomic analysis with STAR, the results will be displayed using topmost gene sets according to the criteria in our response for Major Point 2 of Reviewer 1.

Regarding the consistency between different omic measurements, we believe it is possible to make inferences about pathways or gene sets when multiple -omic measurements agree at multiple levels of biology (metabolites / proteins / RNA). For example, when metabolites, proteins, and transcripts exhibit up-regulation of the same pathway, then we can infer with moderate confidence that the pathway has increased utilization or increased importance. However, the converse is not true. The absence of agreement between different levels of biology (metabolites / proteins / RNA) could be due to genuine biological complexities and/or errors of measurement/analysis. Many biological changes occur at one level of measurement without altering other levels of measurement, such as phosphorylation affecting protein degradation without affecting RNA. Any pathway can have up-regulation of one measure and down-regulation of another due to genuine regulatory mechanisms. Therefore, multi-omic agreement is a positive result, but multi-omics disagreement produces no result, and does not produce a problem or a contradictory result. Previous studies found a correlation < 0.3 between transcriptomic and proteomic levels in the same cells (Haider S and Pal R 2013, Gunawardana Y et al. 2013,Bathke J et al. 2019, Xu JY et al. 2020).Thus, it is no surprise that our data for bat and human ratios (P/W) have a correlation of 0.223 between the proteomic fold-change and the transcriptomic fold-change (see Author response image 3).

**Author response image 3. sa2fig3:** Scatter plot of Log fold change of proteomic data vs Log fold change of transcriptomics data. Each point represents a protein/gene that was detected in both proteomics and transcriptomics experiments. Log fold change was computed in each case as log_2_(PaLung /WI-38).

Reference:

Bathke, J., Konzer, A., Remes, B., McIntosh M., Klug G. Comparative analyses of the variation of the transcriptome and proteome of Rhodobacter sphaeroides throughout growth. BMC Genomics. 2019, 20:1: 358. https://doi.org/10.1186/s12864-019-5749-3.

Haider S, Pal R. Integrated analysis of transcriptomic and proteomic data. Curr Genomics. 2013, 14:2:91-110. doi: 10.2174/1389202911314020003.

Xu JY, Zhang C, Wang X, Zhai L, Ma Y, Mao Y, Qian K, Sun C, Liu Z, Jiang S, Wang M, Feng L, Zhao L, Liu P, Wang B, Zhao X, Xie H, Yang X, Zhao L, Chang Y, Jia J, Wang X, Zhang Y, Wang Y, Yang Y, Wu Z, Yang L, Liu B, Zhao T, Ren S, Sun A, Zhao Y, Ying W, Wang F, Wang G, Zhang Y, Cheng S, Qin J, Qian X, Wang Y, Li J, He F, Xiao T, Tan M. Integrative Proteomic Characterization of Human Lung Adenocarcinoma. Cell 2020, 182:1:245-261.e17. doi: 10.1016/j.cell.2020.05.043.

Gunawardana Y, Niranjan M, Bridging the gap between transcriptome and proteome measurements identifies post-translationally regulated genes, Bioinformatics 2013, 29:23: 3060–3066, https://doi.org/10.1093/bioinformatics/btt537

5. The dominant up-regulation of complex I in ETC is interesting and is the main finding of this paper. However, no experimental evidence was provided to prove the greater activity of Complex I, for example, metabolites changes. In addition, the genes encoding proteins belong to ETC complex I, II, III and IV vary a lot, with much more genes encoding complex I. Therefore, the author should consider the background gene number when they compare the up-regulated gene number differences in each complex. For example, a fisher-exact test could be done to see if complex I has significantly more genes been up-regulated than a random expectation.

We thank the reviewer for raising this issue. This is similar to reviewer 1’s concern about the proportion of ETC complexes raised in major points 7 and 8. As per our response earlier, we do not claim that Complex I genes are upregulated more than other complexes. A chi-square test shows that the proportion of ETC complex genes observed in the core enrichment set of the GSEA gene set is in keeping with the expected frequencies based on the number of genes in each complex. Our claim is only that due to the sheer number of genes in Complex I (even though it is in proportion with expected frequencies), significant enrichment of Complex I alone may be sufficient in most cases to call the entire ETC gene set upregulated, without requiring a strong signal-to-noise ratio of enrichment from the other complexes. We will modify our manuscript to clarify this, and we are grateful for feedback to improve our communication.

Reviewer 2 Minor comments- The author may have to add the p value or FDR for each GSEA plot, even though some of the FDR are not significant. Also, it will be better to show the normalized enrichment score (NES) instead of the ES.

We thank the reviewer for this suggestion and will include the p value/ FDR for the gene sets displayed in the figures.

- The gene set name in several supplementary tables contains many '%' characters and those needs to be removed.

We will amend this in the supplementary material of our revised submission.

- In Line 302, "…combined with the earlier findings of downregulated OxPhos expression and low OCR, we conclude…". If my understanding is right, the authors only mentioned the up-regulation of Oxphos expression, instead of down-regulation. This sentence may need to be clarified.

We will amend this text in our revised submission.

Reviewer 3 Major comments1. The authors state:"We then set the lower bound of the PaLung Complex I reaction flux to a value equal to 70% of its theoretical maximum. Similarly, we set the upper bound of the WI-38 Complex I reaction at a value equal to 30% of its theoretical maximum value. This ensured that the PaLung model would have higher flux through the Complex I reaction, in comparison to the WI-38 model."How do the results hold with different thresholds ? Are these findings robust with e.g. in ranges between 10 to 50% (90-50%) (instead of only 30% and 70%). Furthermore, the histogram figures doesnt seem to reflect a 70% of maximum lower bound for complex I (threshold at a value of 30 seems like extremity of tail).

The reviewer raises a good question, whether our simulation results depend on the thresholds we used. We propose to perform additional flux sampling with thresholds proposed by the reviewer in the range of 10-50%. To help readers interpret the modelling already shown, we will point out that the output flux histograms are the result of combined constraints on Complex I and oxygen intake, so the feasible ranges can be narrower than the flux constraint on Complex I. For the simulations suggested by the reviewer, we warn that using the suggested 10-50% thresholds might generate a conflict with oxygen intake rates and could result in infeasible scenarios. (In that case, the set of possible flux configurations would be the empty set, and would not be plotted in the revised manuscript.)

2. Number of differentially expressed genes is extremely high because such cutoffs are not really meaningful given the comparison between two organisms. No need to refer to the 6247 above cutoff as differentially regulated genes (see: https://elevanth.org/blog/2023/07/17/none-of-the-above/ and https://daniel-saunders-phil.github.io/imagination_machine/posts/if-none-of-the-above-then-what/ for pointers toward current best practice in biological statistics). Enough to simply note that 6247 are above the cutoffs, which suggest a drastic (and expected) difference in expression profiles between the two organisms.

We thank the reviewer for this suggestion. We will refrain from saying “differentially expressed” when comparing different species, and will instead refer to the differences as genes that pass the cutoff threshold.

3. Please highlight the RNA and proteomic analysis assumption and present results within those boundaries (e.g. how are the transcript matched between human and bat, the use of human gene ontologies, etc…). Are the human GO set definitions relevant in bat (it is a common practice with mice and rats, are bats close ?)?

We thank the reviewer for raising this concern. Unlike well studied organisms like human and mice, few bioinformatics resources are readily available for *P. alecto*. Hence in certain steps in our analyses, we used human-derived resources for inter-species comparison. In our revised manuscript, we will reiterate these methodology choices in the discussion :

Use of Human GO terms and gene set definitions for inter-species GSEAUsing only transcripts found in both organisms for transcriptomic analysesMapping of proteomics peptides to human and P. alecto proteins.Use of human derived metabolic mitochondrial models for flux sampling (Mitocore)

4. Are oxphos and hypoxia responses the most extreme pathway scores in the GSEA ? Instead of barcode plots that are generally not a very useful use of figure space, use Figure 1C to show the top e.g.20 (positive and negative) pathway scores so that we can see how much those two actually stand out. Same for the proteomic analysis. Also, need to show an unbiased side by side comparison of the pathway enrichments for RNA and proteomic, the reported results in main text and figures are too cherry picked to be of interest as they stand.

We prefer to provide the full table in supplementary materials because truncating after the top 20 entries would yield too many transcriptomic differences that are irrelevant to metabolism or mitochondria, and therefore impossible to corroborate with the mitochondrial proteomic data. The current figures will be enhanced by adding p-values, but we prefer not to delete the GSEA plots because the p-values don’t provide enough information about the strength of evidence.

5. Finally, and very importantly, please upload ALL the code used for the analysis, with instructions to run it and all the required inputs and source files. The computational analysis is only as credible as it is easy to reproduce.

We apologize if the reviewer found difficulties in running our code (which was uploaded at https://github.com/nsuhasj/PalungWI38FluxSim). We did identify an oversight in the upload, which is the need for better documentation about installing externally available code libraries. We will also perform a blind download on a fresh workstation to confirm that our github repository provides all necessary supporting files.

Reviewer 3 Minor commentsIntroduce GeTMM, what are its key specificities ?

We propose to add more details about GeTMM in our revised manuscript.

Figure 1C code bar plot useless, simply report ES and NES and pathway absolute rank in text.

We acknowledge the reviewer’s comment, but as mentioned in our earlier response, we would prefer to retain the bar plot over the list of gene sets with ES/NES.

Report Foldchange/p-value/rank of complex-I members and other genes of interest for the narrative of the paper.

We will add supplemental material indicating these values.

Reviewer 2 Major comments3. All conclusions are based on high-throughput data, however it is accepted that some validation should be provided. Please provide qPCR or WB (if good antibodies are available) validation for several most significantly differentially expressed genes supporting the pathways identified in Figure 2 (preferably supporting the conclusions about Complexes I/II).

We appreciate the reviewer's valuable suggestions. As the reviewer is aware, it is challenging to find human antibodies to detect bat proteins. Furthermore, for a fair comparison, human antibodies should detect bat proteins with the same affinity as they do human proteins. We aligned the Complex I proteins from humans and bats and found that their homology was relatively low, which suggests that it would be difficult to validate our findings using Western blotting.

Unfortunately, the same challenges apply to validation by quantitative PCR (qPCR). Comparing gene expression by qPCR between different species can be difficult due to several factors, including genetic differences, primer specificity, and the choice of reference genes.

Different species can have variations in their gene sequences, which can affect the binding of qPCR primers and lead to differences in amplification efficiency. Therefore, primers must be designed to target conserved regions of the gene of interest to ensure specificity. However, finding such regions that are conserved between species can be difficult when the gene sequences are not well conserved. In addition, even if the primer annealing region is conserved, amplification efficiency may not be the same if the sequences of the entire amplicon have some variations. As mentioned above, the sequences of the genes in Complex I between humans and bats are not very well conserved.

Nevertheless, we selected eight genes in Complex I that were upregulated in PaLung bat cell lines in our RNAseq or proteomics analysis of mitochondrial proteins, and tested the primers. These genes were NDUFA3, NDUFA7, NDUFA10, NDUFA13, NDUFB2, NDUFB9, NDUFS2, and NDUFV2. The primer sequences were 100% matched to human sequences, but had one or two mismatches to bat sequences. Although we were able to amplify the human fragments well, we were unable to amplify the bat fragments of these genes efficiently or at all. Furthermore, we could not amplify the GAPDH fragments for both human and bat cell lines at comparable levels, even though the primer sequences for GAPDH were 100% matched to both human and bat sequences. Possible explanations for this include variation in GAPDH expression levels between the cell lines, differences in GAPDH amplification efficiency during PCR, or a combination of both factors, even though GAPDH is one of the most commonly used reference genes for qPCR analysis.

In this situation, we believe that RNAseq provides a fair comparison between two different species, as it represents the percentage of expression of a given gene relative to the total mRNA expression. Therefore, rather than attempting to evaluate our findings by Western blot or qPCR, we decided to moderate our conclusions and avoid over-interpreting the data, as suggested by Reviewer 1 in Major Comment 9.

6. If the main findings of this paper can be further confirmed by additional experiments or data, it will be a very nice paper. This could be a potential mechanism that bats used to switch metabolism modes between two metabolic extremes: flight and hibernation, which require high and low energy. However, the usage of only the lung fibroblasts of human and bat may limit the ability of generalizing this 'ischemic-like state' of ETC in most of the bats tissue/organs. While I agree what the authors mentioned in the Discussion section, that to extend to primary cells of other species can help generalize this finding, studying the metabolism state of different cell type of bats (e.g., muscle cells responsible for flight; myocytes and neurons for hibernation) probably can provide more insights into the evolution of various interesting phenotypes of bats.

We would like to thank the reviewer for the insightful comments. We agree that extending the current study to different bat cell types, such as myocytes and neurons, could provide a more comprehensive understanding of metabolic adaptations in bat. This valuable suggestion is beyond the capacity of our laboratory resources and the timeline of this work (i.e., acquisition of primary tissues and generation of cell lines). Our current study provides the first comprehensive comparison of bat and human, to generate interest for subsequent work. We sincerely apologise for this limitation and appreciate the reviewer's understanding. We will include an additional discussion indicating that our work, characterizing differences in TCA cycle, OxPhos and ROS responses in PaLung, acts as a stepping stone for further research that can deepen our understanding of bat metabolism and physiology.

Reviewer 2 Minor comments

- How did mitochondrial DNA content per cell compared between the two species? Could the results be affected by the number and size of the mitochondria per cell in each species? An indirect measurement of mitochondrial DNA yield in the fractionation experiment would be the total DNA amount that was obtained in mitochondrial fractions per cell lysed.

We appreciate the reviewer’s feedback and thoughtful suggestions. The questions about mitochondrial DNA content per cell, and the potential impact of mitochondrial number and size, are indeed relevant to studies of mitochondrial function and metabolism. However, we believe these measurements may not be essential in the context of our study. Regarding the comparison of mitochondrial DNA content between the two species, the genes we observed to be differentially expressed in the mitochondria of bats and humans were encoded by nuclear DNA. As such, measuring mitochondrial DNA content would not significantly alter our main findings.

We acknowledge that our results could potentially be affected by mitochondrial morphology, including the number, size, and fragmentation status of mitochondria. One common approach to evaluating mitochondrial morphology is the use of fluorescence dyes specific to mitochondria, in addition to electron microscopy. However, we found that PaLung bat cells can export most of the staining dyes used to visualize mitochondria, which makes it difficult to accurately assess these parameters. The strong export activity in PaLung cells is due to the high expression of ABCB1 transporter (Koh *et al.*, Nat Commun 2019).

In summary, dyes for assessing mitochondrial morphology have had different behaviors and affinities across species, so we do not wish to assert any conclusions that may be generated through these techniques. We request your understanding that such technical difficulties have prevented us from performing a morphological evaluation of mitochondria to further elucidate our findings. We will include this in the Discussion section.

Reference:

Koh J, Itahana Y, Mendenhall IH, Low D, Soh EXY, Guo AK, Chionh YT, Wang LF, Itahana K. ABCB1 protects bat cells from DNA damage induced by genotoxic compounds. Nat Commun. 2019 Jun 27;10(1):2820. doi: 10.1038/s41467-019-10495-4.

Reviewer Comments1) The code and source data availability are still lack luster. None of the required files to run the only script that they provide are available as of now, neither are instructions on how to run it.

We agree that our first Github upload was insufficient. We have now updated the manuscript with the link to a new Github repository. This repository contains additional files (both data files and scripts) and a README file with instructions on how to run the scripts (and install necessary software). We have also verified that it is possible to run the scripts by performing the installation on a fresh workstation. Our contact information is provided at the repository, and we welcome feedback, including anonymous email.

2) They didn't address my concern about figure 2B constraint looking inconsistent between what is said in method and what is shown on the plot.

We would first like to clarify if the Reviewer might be referring to Figure 3C and not 2B? Figure 2B is a proteomic heatmap and does not involve any constraints.

We interpret the reviewer’s question as an extension of the following query from the original reviewer comments at ReviewCommons

“The authors state:"We then set the lower bound of the PaLung Complex I reaction flux to a value equal to 70% of its theoretical maximum. Similarly, we set the upper bound of the WI-38 Complex I reaction at a value equal to 30% of its theoretical maximum value. This ensured that the PaLung model would have higher flux through the Complex I reaction, in comparison to the WI-38 model."How do the results hold with different thresholds ? Are these findings robust with e.g. in ranges between 10 to 50% (90-50%) (instead of only 30% and 70%). Furthermore, the histogram figures doesn’t seem to reflect a 70% of maximum lower bound for complex I (threshold at a value of 30 seems like extremity of tail).”

In our ReviewCommons response, we said, “To help readers interpret the modelling already shown, we will point out that the output flux histograms are the result of combined constraints on Complex I and oxygen intake, so the feasible ranges can be narrower than the flux constraint on Complex I.” That statement may have been insufficient, so we explain in more detail below.

The histograms in figure 3C show the feasible flux ranges for Complex I and Oxygen intake reactions, after constraints have been placed on both reactions**.** Because of the highly interconnected nature of metabolic networks, placing constraints on one reaction might indirectly limit the feasible flux space of another reaction, which is what we observe here in the case of Complex I (CI) and Mitochondrial Oxygen transport (O2) reactions. To illustrate this, we performed flux sampling simulations under four conditions:

Unconstrained control where both CI and O2 reactions can take any feasible flux value.Where only CI fluxes are constrained to be 30-70, *i.e.,* the CI flux of the bat model has its lower bound set at 70% of its maximum possible value, while that of the human model has its upper bound set at 30% of its maximum possible value.Where only O2 fluxes are constrained to be 30-70, *i.e.,* the O2 flux of the bat model has an upper bound set at 30% of its maximum possible value, while that of the human model has its lower bound set at 70% of its maximum possible value.Fully constrained model where both reactions have been constrained as in Figure 3C of the main manuscript.

It can be seen from Author response image 4 that constraining only CI or only the O2 reaction flux also automatically constrains the other reaction (more p`ronounced in the bat model).

**Author response image 4. sa2fig4:** Flux histograms showing the feasible flux distributions for the Complex I (CI) and the mitochondrial oxygen transport (O2) reactions in the P (PaLung) and W (WI-38) metabolic models. Each column corresponds to setting constraints as per one of the four experiments described above.

In our updated manuscript, we also include Appendix 3 to illustrate the setting of constraints in more detail.

Reviewer:3) As of now, the narrative they present for the choice of focus on the two metabolite pathways they consider from the RNA and proteomic GSEA analysis does not hold. The two pathways they focus on are blended in the middle of hundreds of other pathways that have as much or more evidence of deregulation, but they do not provide any rationale as to why they specifically chose to focus on these two.

Thank you for the opportunity to explain. Below is a detailed explanation of how we chose to focus on the pathways displayed in our main text figures 1 and 2.

Transcriptomics GSEA (Figure 1)

It is true that the two pathways we display “Respiratory electron transport…” and “Cellular response to hypoxia” are not the top differentially regulated pathways and are blended among other altered pathways. Because we seek metabolic differences, we chose gene sets for display as follows:

We first looked only at pathways upregulated in the PaLung samples. This is because pathways downregulated in PaLung could be artifacts due to incomplete/partial annotation of the PaLung genome compared to the human genome. Our updated Discussion section explains this rationale.We then selected gene sets that satisfy the FDR < 0.25 threshold commonly used for GSEA analysis. Justification for why a higher FDR threshold is used for GSEA can be found on the GSEA FAQ website at the following URL https://software.broadinstitute.org/cancer/software/gsea/wiki/index.php/FAQNext, we selected gene sets that had a nominal enrichment score with absolute value >= 1Finally, we manually searched this truncated list of gene sets for pathways related to metabolism, especially “primary metabolism” (central carbon metabolism).

The unfiltered GSEA list is provided as a supplement, and Author response table 1 is a list of the 136 gene sets that passed the first 3 steps of the filtering process, prior to any manual search. We have highlighted in orange the 21 gene sets that we identified as most relevant to metabolism. Note that most of these 21 metabolism-relevant gene sets are NOT part of central carbon metabolism.

**Author response table 1. sa2table1:** 

NAME	SIZE	NES	FDR
L13A-MEDIATED TRANSLATIONAL SILENCING OF CERULOPLASMIN EXPRESSION REACTOME DATABASE ID RELEASE 71 156827	102	-1.86459	0
CAP-DEPENDENT TRANSLATION INITIATION REACTOME DATABASE ID RELEASE 71 72737	110	-1.85211	0
TRANSLATIONAL INITIATION GOBP GO:0006413	114	-1.8446	0
REGULATION OF EXPRESSION OF SLITS AND ROBOS REACTOME DATABASE ID RELEASE 71 9010553	153	-1.84266	0
NONSENSE MEDIATED DECAY (NMD) ENHANCED BY THE EXON JUNCTION COMPLEX (EJC) REACTOME R-HSA-975957.1	107	-1.83796	0
GTP HYDROLYSIS AND JOINING OF THE 60S RIBOSOMAL SUBUNIT REACTOME R-HSA-72706.2	103	-1.82911	0
NUCLEAR-TRANSCRIBED MRNA CATABOLIC PROCESS GOBP GO:0000956	179	-1.82134	0
MRNA CATABOLIC PROCESS GOBP GO:0006402	191	-1.81824	0
EUKARYOTIC TRANSLATION INITIATION REACTOME DATABASE ID RELEASE 71 72613	110	-1.81618	0
EUKARYOTIC TRANSLATION ELONGATION REACTOME R-HSA-156842.2	85	-1.81611	0
SRP-DEPENDENT COTRANSLATIONAL PROTEIN TARGETING TO MEMBRANE REACTOME R-HSA-1799339.2	103	-1.81578	0
SELENOCYSTEINE SYNTHESIS REACTOME DATABASE ID RELEASE 71 2408557	84	-1.81445	0
NONSENSE-MEDIATED DECAY (NMD) REACTOME R-HSA-927802.2	107	-1.81251	0
TRANSLATION REACTOME DATABASE ID RELEASE 71 72766	278	-1.8121	0
RNA CATABOLIC PROCESS GOBP GO:0006401	214	-1.81042	0
PEPTIDE BIOSYNTHETIC PROCESS GOBP GO:0043043	313	-1.8098	0
FORMATION OF A POOL OF FREE 40S SUBUNITS REACTOME DATABASE ID RELEASE 71 72689	92	-1.8098	0
INFLUENZA LIFE CYCLE REACTOME DATABASE ID RELEASE 71 168255	129	-1.80935	0
RESPONSE OF EIF2AK4 (GCN2) TO AMINO ACID DEFICIENCY REACTOME DATABASE ID RELEASE 71 9633012	92	-1.80856	0
VIRAL MRNA TRANSLATION REACTOME DATABASE ID RELEASE 71 192823	81	-1.80841	0
PEPTIDE CHAIN ELONGATION REACTOME R-HSA-156902.2	81	-1.80743	0
NONSENSE MEDIATED DECAY (NMD) INDEPENDENT OF THE EXON JUNCTION COMPLEX (EJC) REACTOME R-HSA-975956.1	87	-1.80499	0
INFLUENZA VIRAL RNA TRANSCRIPTION AND REPLICATION REACTOME DATABASE ID RELEASE 71 168273	121	-1.80381	0
PROTEIN LOCALIZATION TO ENDOPLASMIC RETICULUM GOBP GO:0070972	119	-1.80193	0
NUCLEAR-TRANSCRIBED MRNA CATABOLIC PROCESS, NONSENSE-MEDIATED DECAY GOBP GO:0000184	107	-1.79958	0
SIGNALING BY ROBO RECEPTORS REACTOME R-HSA-376176.4	193	-1.79657	0
INFLUENZA INFECTION REACTOME R-HSA-168254.2	139	-1.79422	0
TRANSLATION GOBP GO:0006412	296	-1.79232	0
EUKARYOTIC TRANSLATION TERMINATION REACTOME R-HSA-72764.4	85	-1.79057	0
SELENOAMINO ACID METABOLISM REACTOME DATABASE ID RELEASE 71 2408522	105	-1.7747	0
MAJOR PATHWAY OF RRNA PROCESSING IN THE NUCLEOLUS AND CYTOSOL REACTOME R-HSA-6791226.3	170	-1.77321	0
ESTABLISHMENT OF PROTEIN LOCALIZATION TO ENDOPLASMIC RETICULUM GOBP GO:0072599	100	-1.76931	0
COTRANSLATIONAL PROTEIN TARGETING TO MEMBRANE GOBP GO:0006613	90	-1.7682	0
RRNA PROCESSING IN THE NUCLEUS AND CYTOSOL REACTOME R-HSA-8868773.3	180	-1.76363	2.64E-05
PROTEIN TARGETING TO MEMBRANE GOBP GO:0006612	134	-1.76038	2.57E-05
CYTOPLASMIC RIBOSOMAL PROTEINS WIKIPATHWAYS_20191210 WP477 HOMO SAPIENS	82	-1.75783	4.97E-05
PROTEIN TARGETING TO ER GOBP GO:0045047	97	-1.75446	7.25E-05
VIRAL GENE EXPRESSION GOBP GO:0019080	122	-1.75351	7.06E-05
SRP-DEPENDENT COTRANSLATIONAL PROTEIN TARGETING TO MEMBRANE GOBP GO:0006614	85	-1.75031	6.88E-05
VIRAL TRANSCRIPTION GOBP GO:0019083	105	-1.74701	6.71E-05
ACTIVATION OF THE MRNA UPON BINDING OF THE CAP-BINDING COMPLEX AND EIFS, AND SUBSEQUENT BINDING TO 43S REACTOME R-HSA-72662.3	56	-1.74344	8.72E-05
AMIDE BIOSYNTHETIC PROCESS GOBP GO:0043604	383	-1.74084	8.51E-05
NUCLEOBASE-CONTAINING COMPOUND CATABOLIC PROCESS GOBP GO:0034655	322	-1.73596	1.04E-04
CYTOPLASMIC TRANSLATION GOBP GO:0002181	54	-1.73436	1.02E-04
RRNA PROCESSING REACTOME DATABASE ID RELEASE 71 72312	189	-1.73364	9.96E-05
TRANSLATION INITIATION COMPLEX FORMATION REACTOME DATABASE ID RELEASE 71 72649	55	-1.73183	9.75E-05
RIBOSOMAL SCANNING AND START CODON RECOGNITION REACTOME R-HSA-72702.3	55	-1.71426	2.11E-04
PROTEIN TARGETING GOBP GO:0006605	283	-1.70904	2.62E-04
FORMATION OF THE TERNARY COMPLEX, AND SUBSEQUENTLY, THE 43S COMPLEX REACTOME DATABASE ID RELEASE 71 72695	48	-1.70325	4.03E-04
CELLULAR NITROGEN COMPOUND CATABOLIC PROCESS GOBP GO:0044270	345	-1.69784	4.66E-04
ESTABLISHMENT OF PROTEIN LOCALIZATION TO ORGANELLE GOBP GO:0072594	325	-1.69391	6.33E-04
AROMATIC COMPOUND CATABOLIC PROCESS GOBP GO:0019439	347	-1.6925	6.90E-04
PEPTIDE METABOLIC PROCESS GOBP GO:0006518	392	-1.69196	6.94E-04
CELLULAR RESPONSES TO STRESS REACTOME DATABASE ID RELEASE 71 2262752	456	-1.67591	0.001561
HETEROCYCLE CATABOLIC PROCESS GOBP GO:0046700	343	-1.66849	0.00217
ESTABLISHMENT OF PROTEIN LOCALIZATION TO MEMBRANE GOBP GO:0090150	215	-1.66714	0.00226
CELLULAR RESPONSES TO EXTERNAL STIMULI REACTOME DATABASE ID RELEASE 71 8953897	459	-1.66708	0.002236
ORGANIC CYCLIC COMPOUND CATABOLIC PROCESS GOBP GO:1901361	365	-1.64869	0.004763
CALNEXIN CALRETICULIN CYCLE REACTOME R-HSA-901042.2	23	-1.60474	0.022972
N-GLYCAN TRIMMING IN THE ER AND CALNEXIN CALRETICULIN CYCLE REACTOME DATABASE ID RELEASE 71 532668	32	-1.59981	0.026341
RIBOSOMAL LARGE SUBUNIT BIOGENESIS GOBP GO:0042273	64	-1.58547	0.041181
OXYGEN-DEPENDENT PROLINE HYDROXYLATION OF HYPOXIA-INDUCIBLE FACTOR Α REACTOME DATABASE ID RELEASE 71 1234176	61	-1.58214	0.045084
ER QUALITY CONTROL COMPARTMENT (ERQC) REACTOME DATABASE ID RELEASE 71 901032	18	-1.57559	0.054216
RIBOSOME ASSEMBLY GOBP GO:0042255	49	-1.56563	0.071338
AMINO ACID AND DERIVATIVE METABOLISM REACTOME R-HSA-71291.6	282	-1.56487	0.071759
CITRIC ACID CYCLE (TCA CYCLE) REACTOME DATABASE ID RELEASE 71 71403	22	-1.56128	0.078243
TRANSLATION FACTORS WIKIPATHWAYS_20191210 WP107 HOMO SAPIENS	48	-1.55985	0.079794
REGULATION OF TP53 DEGRADATION REACTOME R-HSA-6804757.1	31	-1.55649	0.086393
VIRAL PROCESS GOBP GO:0016032	259	-1.5553	0.08777
SIGNALING BY FGFR4 REACTOME R-HSA-5654743.2	27	-1.55424	0.088823
REGULATION OF CALCIUM-MEDIATED SIGNALING GOBP GO:0050848	47	-1.55084	0.095609
NEGATIVE REGULATION OF G0 TO G1 TRANSITION GOBP GO:0070317	36	-1.54615	0.106841
CYCLIN D ASSOCIATED EVENTS IN G1 REACTOME R-HSA-69231.7	40	-1.54534	0.107785
G1 PHASE REACTOME DATABASE ID RELEASE 71 69236	40	-1.54481	0.107951
INSULIN PROCESSING REACTOME R-HSA-264876.2	20	-1.54226	0.114012
ERROR-PRONE TRANSLESION SYNTHESIS GOBP GO:0042276	19	-1.53741	0.126991
TRANSLESION SYNTHESIS BY POLH REACTOME R-HSA-110320.1	18	-1.53506	0.132554
SMOOTH MUSCLE CONTRACTION REACTOME R-HSA-445355.3	29	-1.53357	0.136112
CELLULAR RESPONSE TO HYPOXIA REACTOME R-HSA-1234174.2	69	-1.53084	0.143644
INTERSPECIES INTERACTION BETWEEN ORGANISMS GOBP GO:0044419	322	-1.53011	0.144136
DUAL INCISION IN TC-NER REACTOME R-HSA-6782135.1	62	-1.53011	0.142368
TCA CYCLE (AKA KREBS OR CITRIC ACID CYCLE) WIKIPATHWAYS_20191210 WP78 HOMO SAPIENS	18	-1.52732	0.149964
SYMBIONT PROCESS GOBP GO:0044403	316	-1.52634	0.151093
REGULATION OF TP53 EXPRESSION AND DEGRADATION REACTOME R-HSA-6806003.1	32	-1.5262	0.149924
ERROR-FREE TRANSLESION SYNTHESIS GOBP GO:0070987	19	-1.52572	0.149766
TRANSCRIPTIONAL REGULATION BY E2F6 REACTOME DATABASE ID RELEASE 71 8953750	34	-1.52515	0.150193
MITOPHAGY REACTOME DATABASE ID RELEASE 71 5205647	25	-1.52402	0.152274
GAP-FILLING DNA REPAIR SYNTHESIS AND LIGATION IN TC-NER REACTOME DATABASE ID RELEASE 71 6782210	62	-1.52125	0.161169
ER-PHAGOSOME PATHWAY REACTOME DATABASE ID RELEASE 71 1236974	69	-1.52107	0.159943
SYNTHESIS OF ACTIVE UBIQUITIN: ROLES OF E1 AND E2 ENZYMES REACTOME R-HSA-8866652.2	29	-1.52027	0.160681
FCERI MEDIATED NF-ΚB ACTIVATION REACTOME R-HSA-2871837.2	71	-1.51881	0.164231
SIGNALING BY *Fgfr1* REACTOME DATABASE ID RELEASE 71 5654736	32	-1.51836	0.164082
CENTRAL NERVOUS SYSTEM NEURON DEVELOPMENT GOBP GO:0021954	29	-1.51808	0.163292
REGULATION OF TP53 ACTIVITY THROUGH METHYLATION REACTOME DATABASE ID RELEASE 71 6804760	17	-1.51684	0.166234
AUF1 (HNRNP D0) BINDS AND DESTABILIZES MRNA REACTOME DATABASE ID RELEASE 71 450408	50	-1.51593	0.167798
B CELL ACTIVATION REACTOME DATABASE ID RELEASE 71 983705	92	-1.5138	0.173984
CYTOPLASMIC PATTERN RECOGNITION RECEPTOR SIGNALING PATHWAY GOBP GO:0002753	32	-1.51365	0.172791
IKK COMPLEX RECRUITMENT MEDIATED BY RIP1 REACTOME DATABASE ID RELEASE 71 937041	18	-1.51332	0.172421
VIRION ASSEMBLY GOBP GO:0019068	35	-1.51249	0.173743
ENDOSOMAL SORTING COMPLEX REQUIRED FOR TRANSPORT (ESCRT) REACTOME R-HSA-917729.1	27	-1.5117	0.174884
INFECTIOUS DISEASE REACTOME DATABASE ID RELEASE 71 5663205	379	-1.51083	0.176457
CHONDROITIN SULFATE METABOLIC PROCESS GOBP GO:0030204	26	-1.50909	0.181727
PROTEIN LOCALIZATION TO MEMBRANE GOBP GO:0072657	357	-1.5088	0.18113
I-KAPPAB KINASE/NF-KAPPAB SIGNALING GOBP GO:0007249	51	-1.50817	0.181788
TICAM1, RIP1-MEDIATED IKK COMPLEX RECRUITMENT REACTOME R-HSA-168927.3	17	-1.50722	0.183755
RIBOSOME BIOGENESIS GOBP GO:0042254	227	-1.50672	0.183798
VESICLE-MEDIATED TRANSPORT BETWEEN ENDOSOMAL COMPARTMENTS GOBP GO:0098927	21	-1.50447	0.190799
DEGRADATION OF *GLI2* BY THE PROTEASOME REACTOME R-HSA-5610783.1	56	-1.50387	0.191333
CITRATE METABOLIC PROCESS GOBP GO:0006101	29	-1.50382	0.189784
ATP METABOLIC PROCESS GOBP GO:0046034	136	-1.50016	0.202786
PRADER-WILLI AND ANGELMAN SYNDROME WIKIPATHWAYS_20191210 WP3998 HOMO SAPIENS	31	-1.49851	0.20778
MYD88-INDEPENDENT TOLL-LIKE RECEPTOR SIGNALING PATHWAY GOBP GO:0002756	27	-1.49841	0.206341
ANTIGEN PROCESSING AND PRESENTATION OF PEPTIDE ANTIGEN VIA MHC CLASS I GOBP GO:0002474	80	-1.49762	0.207835
STABILIZATION OF P53 REACTOME R-HSA-69541.5	53	-1.49515	0.216427
DOWNSTREAM TCR SIGNALING REACTOME R-HSA-202424.3	76	-1.49442	0.217854
THE ROLE OF GTSE1 IN G2 M PROGRESSION AFTER G2 CHECKPOINT REACTOME DATABASE ID RELEASE 71 8852276	57	-1.49062	0.233181
ABC TRANSPORTER DISORDERS REACTOME DATABASE ID RELEASE 71 5619084	64	-1.49037	0.232277
GAP-FILLING DNA REPAIR SYNTHESIS AND LIGATION IN GG-NER REACTOME R-HSA-5696397.1	24	-1.48948	0.234394
TNFR2 NON-CANONICAL NF-ΚB PATHWAY REACTOME R-HSA-5668541.2	80	-1.48934	0.233057
CLEC7A (DECTIN-1) SIGNALING REACTOME R-HSA-5607764.1	89	-1.4891	0.232317
REGULATION OF G0 TO G1 TRANSITION GOBP GO:0070316	38	-1.48792	0.235624
NEGATIVE REGULATION OF FGFR4 SIGNALING REACTOME DATABASE ID RELEASE 71 5654733	19	-1.48738	0.236268
VIF-MEDIATED DEGRADATION OF APOBEC3G REACTOME DATABASE ID RELEASE 71 180585	50	-1.48711	0.235552
G1 S DNA DAMAGE CHECKPOINTS REACTOME R-HSA-69615.2	63	-1.48708	0.233782
GLI3 IS PROCESSED TO GLI3R BY THE PROTEASOME REACTOME R-HSA-5610785.1	56	-1.48693	0.232652
RESPIRATORY ELECTRON TRANSPORT, ATP SYNTHESIS BY CHEMIOSMOTIC COUPLING, AND HEAT PRODUCTION BY UNCOUPLING PROTEINS. REACTOME R-HSA-163200.1	107	-1.48672	0.231731
REGULATION OF TP53 ACTIVITY THROUGH PHOSPHORYLATION REACTOME DATABASE ID RELEASE 71 6804756	85	-1.48642	0.231313
BUDDING AND MATURATION OF HIV VIRION REACTOME DATABASE ID RELEASE 71 162588	24	-1.48594	0.231623
SIGNALING BY FGFR REACTOME R-HSA-190236.2	61	-1.48574	0.230753
PHOSPHODIESTERASES IN NEURONAL FUNCTION WIKIPATHWAYS_20191210 WP4222 HOMO SAPIENS	29	-1.48396	0.237066
P53-DEPENDENT G1 DNA DAMAGE RESPONSE REACTOME DATABASE ID RELEASE 71 69563	61	-1.48333	0.237846
PINK PARKIN MEDIATED MITOPHAGY REACTOME R-HSA-5205685.2	20	-1.48228	0.24089
MFAP5 EFFECT ON PERMEABILITY AND MOTILITY OF ENDOTHELIAL CELLS VIA CYTOSKELETON REARRANGEMENT WIKIPATHWAYS_20191210 WP4560 HOMO SAPIENS	16	-1.48205	0.240132
FC EPSILON RECEPTOR (FCERI) SIGNALING REACTOME DATABASE ID RELEASE 71 2454202	108	-1.48203	0.238453
OXIDATIVE STRESS INDUCED SENESCENCE REACTOME R-HSA-2559580.4	65	-1.48099	0.241159

Only 21 gene sets pass the filtering for relevance and significance (from a starting count of 18000+ gene sets). Of these 21 gene sets, 5 pathways related to TCA cycle/electron transport chain (ETC) and 3 gene sets related to hypoxia and stress response. Many of the other pathways pertain to anabolism/catabolism or secondary metabolism (e.g., RNA catabolism, translation, amide metabolic process, seleno amino acid metabolism, chondroitin sulfate metabolism etc). This is why we chose to highlight the TCA/ETC gene set and the hypoxia gene set in Figure 1.

Although the selected gene sets are not the topmost altered metabolic pathways in our list, they are still significantly altered, according to FDR and NES. Furthermore, by belonging to central carbon metabolism, they provide greater insight into fundamental metabolic differences between the two species compared.

Our manuscript now includes the following text in the transcriptomics result section.

“Since the *P. alecto* genome is less fully annotated than the human genome, pathways with incomplete annotation may be incorrectly predicted to be downregulated in PaLung cells. Hence, we only studied differentially regulated pathways that were upregulated in PaLung. When we filtered PaLung-upregulated gene sets for significance (indicated by FDR<0.25 and normalized enrichment score |NES| > 1), and for relevance to metabolism, only 21 gene sets remained. Many were relevant to secondary metabolism or anabolic/catabolic housekeeping, five were related to the TCA cycle and electron transport (including “ATP synthesis by chemiosmotic coupling, respiratory electron transport, and heat production by uncoupling proteins”) and three were related to hypoxic stress (such as “Cellular response to hypoxia” in the Reactome Pathway Database).”

Proteomics GSEA (Figure 2)

Following similar logic as with the transcriptomics GSEA (upregulated in PaLung, FDR < 0.25, |NES| > = 1), resulted in a total of 118 gene sets. The top 20 gene sets in this list (Author response table 2) contained 8 gene sets that pertained to metabolism (highlighted in orange).

**Author response table 2. sa2table2:** 

NAME	SIZE	NES	FDR
MUSCLE CONTRACTION GOBP GO:0006936	35	-2.29	0.008034
NICOTINIC ACETYLCHOLINE RECEPTOR SIGNALING PATHWAY PANTHER PATHWAY P00044	15	-2.17947	0.018238
REGULATION OF CELL JUNCTION ASSEMBLY GOBP GO:1901888	15	-2.17782	0.012159
THE CITRIC ACID (TCA) CYCLE AND RESPIRATORY ELECTRON TRANSPORT REACTOME R-HSA-1428517.1	64	-2.13697	0.015138
HALLMARK_OXIDATIVE_PHOSPHORYLATION MSIGDB_C2 HALLMARK_OXIDATIVE_PHOSPHORYLATION	91	-2.12375	0.013852
COLLAGEN FORMATION REACTOME DATABASE ID RELEASE 71 1474290	31	-2.11295	0.013374
MUSCLE SYSTEM PROCESS GOBP GO:0003012	40	-2.07522	0.017172
EPH-EPHRIN SIGNALING REACTOME DATABASE ID RELEASE 71 2682334	30	-2.06553	0.016753
ACTIN FILAMENT-BASED MOVEMENT GOBP GO:0030048	17	-2.05773	0.017006
RESPIRATORY ELECTRON TRANSPORT REACTOME R-HSA-611105.3	32	-2.0515	0.016612
RHO GTPASES ACTIVATE PKNS REACTOME DATABASE ID RELEASE 71 5625740	17	-2.0504	0.015369
COLLAGEN BIOSYNTHESIS AND MODIFYING ENZYMES REACTOME R-HSA-1650814.3	25	-2.03023	0.018264
RESPIRATORY ELECTRON TRANSPORT, ATP SYNTHESIS BY CHEMIOSMOTIC COUPLING, AND HEAT PRODUCTION BY UNCOUPLING PROTEINS. REACTOME R-HSA-163200.1	32	-2.0253	0.017967
COMPLEX I BIOGENESIS REACTOME R-HSA-6799198.1	16	-2.02379	0.016892
MITOCHONDRIAL RESPIRATORY CHAIN COMPLEX I ASSEMBLY GOBP GO:0032981	16	-2.01705	0.01673
INTEGRIN SIGNALLING PATHWAY PANTHER PATHWAY P00034	44	-2.00758	0.017239
NADH DEHYDROGENASE COMPLEX ASSEMBLY GOBP GO:0010257	16	-1.99919	0.017902
MITOCHONDRIAL ATP SYNTHESIS COUPLED ELECTRON TRANSPORT GOBP GO:0042775	26	-1.99497	0.01756
ACTOMYOSIN STRUCTURE ORGANIZATION GOBP GO:0031032	21	-1.98512	0.018578
KERATINIZATION GOBP GO:0031424	15	-1.96272	0.024099

All 8 pathways belonged to the TCA cycle/ETC, within which 3 correspond to Complex I of ETC, which is why we display these gene sets in Figure 2 of the main text.

An additional response to the Reviewer’s question is also now provided in Appendix 2**,** which addresses Reviewer 1’s request (Major comment 6). This supplement re-computes our proteomics GSEA using a mitochondria-only geneset list (Mitocarta 3.0; URL: https://www.broadinstitute.org/mitocarta/mitocarta30-inventory-mammalian-mitochondrial-proteins-and-pathways), instead of the full GO BP gene set list. Results from this new GSEA show that OxPhos is the top upregulated pathway in the bat samples. Complex I is also among the top 5 gene sets found upregulated in bat proteomic samples. Author response table 3 shows all the gene sets upregulated in the bat samples, with FDR value < 0.25.

**Author response table 3. sa2table3:** Gene sets enriched in phenotype P (3 samples).

	GS follow link to MSigDB	GS DETAILS	SIZE	ES	NES	NOM p-val	FOR q-val	FWER p-val
	OXPHOS SUBUNITS	Details.	34	-0.55	-2.19	o.ooo	0.002	0.001
2	OXPHOS	Details.	41	-0.53	-2.17	o.ooo	0.001	0.001
3	CARBOHYDRATE METABOLISM	Details.	36	-0.50	-2.00	o.ooo	0.005	0.009
4	TRANSLATION	Details.	16	-0.62	-1.99	0.002	0.005	0.011
5	COMPLEX I	Details.	17	-0.60	-1.97	0.004	0.005	0.013
6	Cl SUBUNITS	Details.	15	-0.62	-1.89	0.004	0.007	0.024
7	FATTY ACID OXIDATION	Details.	20	-0.55	-1.86	0.008	0.007	0.030
8	METALS AND COFACTORS	Details.	30	-0.47	1.79	0.006	0.011	0.051
9	METABOLISM	Details.	153	-0.34	1.78	0.002	0.011	0.058
10	AMINO ACID METABOLISM	Details.	33	-0.46	1.76	0.011	0.011	0.066
11	MITOCHONDRIAL CENTRAL DOGMA	Details.		-0.45	-1.61	0.012	0.030	o. 181
12	TCA CYCLE	Details.	15	-0.48	-1.51	0.052	0.053	0.310
13	LIPID METABOLISM	Details.	43	-0.36	-1.48	0.032	0.057	0.359
14	PROTEIN IMPORT SORTING AND HOMEOSTASIS	Details.	24	-0.34	-1.20	0.205	0.223	0.846